# DNA-bridging by an archaeal histone variant via a unique tetramerisation interface

Sapir Ofer[1,5], Fabian Blombach[1,5], Amanda M. Erkelens[2], Declan Barker[1], Zoja Soloviev[1], Samuel Schwab[2], Katherine Smollett[1], Dorota Matelska[1,4], Thomas Fouqueau[1], Nico van der Vis[2], Nicholas A. Kent[3], Konstantinos Thalassinos[1], Remus T. Dame[2✉] & Finn Werner[1✉]

In eukaryotes, histone paralogues form obligate heterodimers such as H3/H4 and H2A/H2B that assemble into octameric nucleosome particles. Archaeal histones are dimeric and assemble on DNA into 'hypernucleosome' particles of varying sizes with each dimer wrapping 30 bp of DNA. These are composed of canonical and variant histone paralogues, but the function of these variants is poorly understood. Here, we characterise the structure and function of the histone paralogue MJ1647 from *Methanocaldococcus jannaschii* that has a unique C-terminal extension enabling homotetramerisation. The 1.9 Å X-ray structure of a dimeric MJ1647 species, structural modelling of the tetramer, and site-directed mutagenesis reveal that the C-terminal tetramerization module consists of two alpha helices in a handshake arrangement. Unlike canonical histones, MJ1647 tetramers can bridge two DNA molecules in vitro. Using single-molecule tethered particle motion and DNA binding assays, we show that MJ1647 tetramers bind ~60 bp DNA and compact DNA in a highly cooperative manner. We furthermore show that MJ1647 effectively competes with the transcription machinery to block access to the promoter in vitro. To the best of our knowledge, MJ1647 is the first histone shown to have DNA bridging properties, which has important implications for genome structure and gene expression in archaea.

[1] Institute for Structural and Molecular Biology, Division of Biosciences, University College London, Darwin Building, Gower Street, London WC1E 6BT, UK. [2] Leiden Institute of Chemistry, Leiden University, Leiden, the Netherlands. [3] School of Biosciences, Cardiff University, Museum Avenue, Cardiff CF10 3AX, UK. [4] Present address: Centre for Genomics Research, Discovery Sciences, BioPharmaceuticals R&D, AstraZeneca, Cambridge, UK. [5] These authors contributed equally: Sapir Ofer, Fabian Blombach. ✉email: rtdame@chem.leidenuniv.nl; f.werner@ucl.ac.uk

Histone-based chromatin is widely present in the Archaea and predates the origin of eukaryotes from an archaeal ancestor[1,2]. Histone fold proteins have also been identified in bacteria[3] and histone-based chromatin has recently been identified in the bacterium *Bdellovibrio bacteriovorus* albeit with a radically different DNA-binding mode compared to archaeoeukaryotic histones[4]. Archaeal histones bear a close structural resemblance to their eukaryotic counterparts while also differing in several important aspects[5]. The archaeal histone encompasses the classical histone fold but lacks N- or C-terminal extensions, and there is no evidence of the extensive post-translational modifications that are characteristic for eukaryotic histones and which regulate transcription[6]. The tertiary structure of archaeal histones is near-identical to their eukaryotic counterparts. Eukaryotic histones H2A, H2B, H3 and H4 form well-characterised octameric nucleosomes that wrap ~147 bp of DNA. Canonical archaeal histones, in contrast, form dimers that assemble on DNA into 'hypernucleosome' particles of varying sizes with each dimer wrapping 30 bp of DNA[5,7]. The X-ray structure encompassing three *Methanothermus fervidus* histone homodimers bound to a 90-bp DNA fragment showed dimensions including the diameter and step size for each solenoid turn identical to those of the eukaryotic nucleosome. However, the archaeal hypernucleosome forms a rod-like protein core around which the DNA is wound in a left-handed solenoid[7]. This arrangement is unlike the eukarytic nucleosome octamer, but it shows similarity to the structure of chromatin formed by telomeric tetranucleosomes in eukaryotes[8]. Analytical ultracentrifugation and cryo-electron microscopy (cryo-EM) experiments suggest that the size of archaeal hypernucleosomes is relatively short[9]. In contrast, Tethered particle motion (TPM) and magnetic tweezer (MT) experiments measuring DNA compaction for two archaeal model histones HMfA and HMfB indicated extended hypernucleosome formation[10]. Limited digestion of archaeal chromatin with Micrococcal Nuclease (MNase) produces a nucleosome ladder with step increments of 30 bp up to >400 bp length[11]. The MNase data are thus consistent with histone homodimers binding to 30 bp DNA as minimal chromatin subunit and forming longer hypernucleosomes in vivo[11,12]. The hypernucleosome size distribution in cells could be affected by different factors. Histone variants with weakened dimer–dimer interfaces might act as 'capstones' that limit hypernucleosome size[13]. The number of stacking interactions available to different histone variants was shown to affect hypernucleosome stability in vitro[5,10]. Lastly, post-translational modification (acetylation) of histone lysines involved in hypernucleosome stacking interactions has been observed in *Thermococcus gammatolerans* and is likely to affect hypernucleosome stability in vivo[14].

Eukaryotic nucleosomes self-interact to form phase-separated aggregates or—under in vitro conditions—structured 30 nm fibres[15]. Higher-order nucleosome organisations are furthermore stabilised by linker histones H1 and H5[5]. Analytical ultracentrifugation experiments suggest in contrast that archaeal hypernucleosomal fibres do not readily associate with each other[9]. These findings gave rise to the idea that the inherent flexibility of archaeal hypernucleosomes is required in the absence of eukaryotic-like chromatin remodelling factors to facilitate transcription of chromatinised DNA[9]. Different archaea encode a plethora of nucleoid-associated proteins (NAPs) in addition to the histones, including small basic proteins such as the highly abundant Alba, Cren7 and Sul7, the latter two being specific to 'histone-free' archaeal species belonging to the crenarchaeota[1]. Most archaea encode several histone paralogues that are closely related on the sequence level with little variation in length, amino acid composition and domain organisation.

Importantly, there is a mutual interference between chromatin formation and biological processes that utilise the genomic DNA as template such as transcription that is thought to modulate hypernucleosome formation[16]. Conversely, chromatin has the potential to deny access of the transcription machinery to gene promoters and thereby to regulate gene expression. Supporting this notion, the deletion or mutation of histones in *Methanosarcina mazei* and *Thermooccus kodakarensis* results in aberrant gene expression patterns[17,18], and a severe impairment of DNA recombination[19]. Notably, the borders between chromatin proteins and transcription regulators are fluid. In *Halobacterium* salinarum, the single histone HypA evolved to act as a transcription regulator that binds to discrete genomic sites rather than playing a role in genome compaction[20,21]. A limited number of studies have characterised the impact of archaeal histones on transcription under rigorously defined conditions in vitro[16,22–24]. These studies emphasise that archaeal histones have an overall attenuating or inhibitory effect on transcription. Transcription elongation factors including the transcript cleavage factor TFS and the processivity factor Spt4/5 enhance the transcription of archaeal RNA polymerase (RNAP) through histone-based chromatin[25]. However, it remains unknown how the combination of histone variants affects transcription.

Members of the Methanococcales encode an unusual histone variant exemplified by MJ1647 from *Methanocaldococcus jannaschii*[26]. MJ1647 homologues harbour a divergent histone fold that is followed by a C-terminal extension of ~27 amino acids.

To shed light on archaeal histones and chromatin in vitro and in vivo, we have applied a multidisciplinary approach to characterise the structure and function of two representative histone variants from *M. jannaschii*, the canonical histone A3 and MJ1647. A combination of X-ray crystallography, structure modelling and molecular genetics shows that MJ1647 is a tetramer formed by two histone dimers that can bridge two DNA molecules. Single-molecule tethered particle motion and biochemical assays combined demonstrate that both A3 and MJ1647 bind cooperatively to and compact DNA, with A3 dimers proceeding in canonical 30 bp-step increments of DNA, while MJ1647 tetramers proceed in 60-bp steps. In vitro transcription experiments show that both A3 and MJ1647 negatively interfere with transcription pre-initiation complex (PIC) recruitment and transcription.

## Results

**MJ1647 variant histones form tetramers via their C-terminal extension**. Most archaea encode multiple histones of ~67 amino acids comprising the canonical triple-helical histone fold that form dimers in solution. *M. jannaschii* encodes four canonical histone paralogues, three on the main chromosome termed A1 to A3 that are highly abundant[6,27], and a less abundant paralogue on the extrachromosomal elements (MJECL29). In addition to these canonical histones, *M. jannaschii* also encompasses the histone variant MJ1647 that is unique to members of the Methanococcales (arCOG02145). MJ1647 includes a 27 amino acid C-terminal extension of the canonical triple-helical histone fold with unknown function. Based on *M. jannaschii* RNA-seq data[28], *mj1647* transcripts comprise ~4% of histone-encoding transcripts and show a similar expression level as the gene encoding Alba. MJ1647 is highly divergent from the four canonical histones (Fig. 1). To test the oligomerisation state, we analysed recombinant A3 and MJ1647 as well as a deletion variant of MJ1647 lacking the C-terminal 27 amino acids by native mass spectrometry as well as Size Exclusion Chromatography followed by Multi-Angle Light Scattering (SEC-MALS). In SEC-MALS, all three proteins eluted as single peaks (Fig. 1b–d). A3 showed an apparent native molecular weight of ~13.4 kDa in agreement with

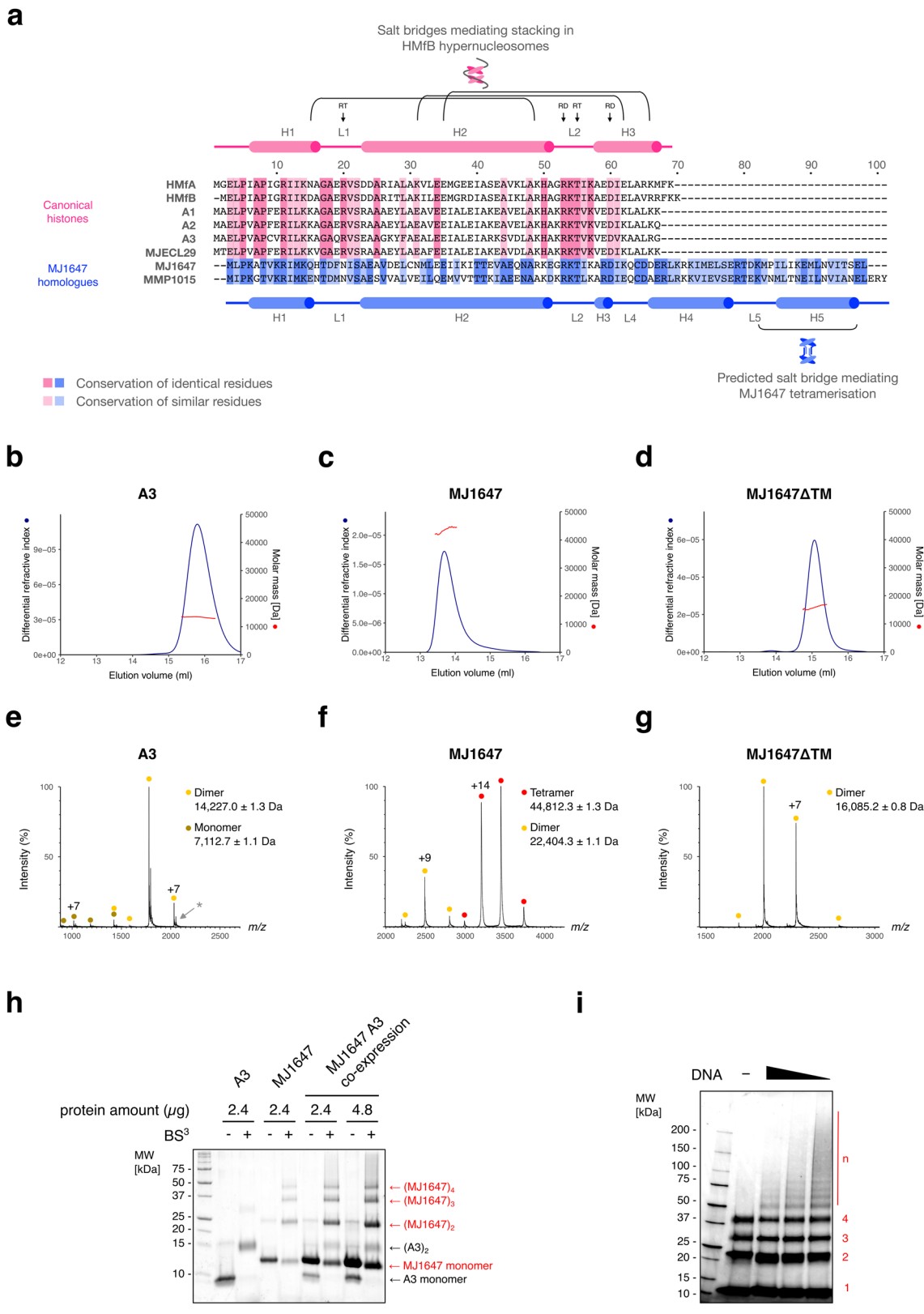

the expected 14.5 kDa for a canonical histone dimer. In contrast, MJ1647 showed an apparent native molecular weight of 43.6 kDa, indicating that MJ1647 forms tetramers in solution (44.8 kDa expected mass). Notably, the C-terminal extension truncation mutant of MJ1647 forms 15.9 kDa dimers akin to the canonical histones (16.1 kDa expected).

To get more accurate mass measurements, we performed native spectrometry experiments for all three proteins. The mass spectra were consistent with MJ1647 forming mainly tetramers in contrast to dimers formed by the MJ1647 C-terminal truncation variant and A3 with a mass error within 1.3 Da for all three proteins (Fig. 1e–g).

**Fig. 1 The C-terminal extension of the MJ1647 variant histone confers dimerisation of dimers. a** Multiple sequence alignment of MJ1647 and its *M. maripaludis* homologue MMP1015 along the four canonical histones from *M. jannaschii* (A1, A2, A3 and MJECL25) and the well-characterised archaeal model histones from *M. fervidus* HMfA and HMfB. Identical and similar residues within each clade are highlighted as indicated based on alignments of 95 canonical histones (arCOG2144) and 14 MJ1647 homologues (arCOG02145). The secondary structure of HmfB and MJ1647 (see further below) is indicated above (HMfB, in pink) and below the alignment (MJ1647, blue). The 'RT'-pair and "RD"-clamp form two conserved pairs of residues mediating DNA-binding by HMfB[7,29,30]. **b–d** SEC-MALS analysis reveals the molecular weight of A3 as ~13.4 kDa congruent with a dimeric histone (**b**), MJ1647 with ~43.6 kDa as tetrameric histone (**b**), and the truncation variant MJ1647ΔTM with ~15.9 kD as dimeric form (**d**). Chromatograms showing the elution profile with x axis: volume (mL), y axes: differential refractive index (left) and molar mass (Da) (right). **e–g** Nano-electrospray ionisation mass spectra of A3 (**e**), MJ1647 (**f**) and MJ1647ΔTM (**g**). Filled circles indicate the charge state series for different oligomerisation states. A3 gave rise to an additional charge state series (marked with a grey asterisk) with a molecular weight difference of 131 Da that can be attributed to partial N-terminal methionine processing during heterologous expression in *E. coli*. **h** BS$^3$ cross-linking of recombinantly co-expressed MJ1647 and A3 compared to control reactions with recombinant MJ1647 and A3 alone. Cross-linked proteins were subjected to SDS-PAGE and stained with SYPRO-Orange. The assignment of the number of A3 and MJ1647 monomers cross-linked is based on the control reactions for A3 and MJ1647 alone. **i** MJ1647 forms higher-order oligomers in the presence of DNA. Histone MJ1647 was subjected to cross-linking with BS$^3$ in the presence of increasing amounts of ~0.5 kb DNA (0.25, 0.5 and 1 µg). Cross-linked samples were resolved by SDS-PAGE stained with protein-stain SYPRO-Orange. The number of cross-linked monomers in each band are indicated in red with 'n' denoting higher-order cross-linked complexes formed in the presence of DNA.

Closely related archaeal histone paralogues are thought to heterodimerise in a promiscuous manner[13] prompting us to test whether the highly divergent A3 and MJ1647 can cross-oligomerise. We tested whether A3 and MJ1647 could form mixed oligomers by co-expressing A3 and MJ1647 in *E. coli* and analysing the complexes by SEC (Fig. 1h). A3 and MJ1647 eluted from the SEC column in two sharp and distinct peaks which correspond to A3 dimers and MJ1647 tetramers according to their apparent molecular weight. This profile mirrored the individually expressed A3 and MJ1647 histones. We could not observe a peak in an intermediate elution volume predicted for A3-MJ1647 hetero-dimers (Supplementary Fig. 1). In addition, we carried out cross-linking experiments with the amine-specific crosslinker bis(sulfo-succinimidyl)suberate (BS$^3$) and analysed the cross-linked species by SDS-PAGE that provides increased resolution compared to SEC. A3 alone resulted in cross-linked dimers. MJ1647 alone resulted in cross-linked dimers, trimer and tetramers. The co-expressed A3 and MJ1647 led to cross-linking products reflecting a combination of A3 and MJ1647 but not new heterodimeric species, predicted to form an additional band between A3 dimers and MJ1647 dimers (Fig. 1h). Previous formaldehyde cross-linking experiments retrieved only MJ1647 dimers, and not tetramers[26]. The BS$^3$ cross-linking with a longer cross-linking distance (~11.4 Å) compared to formaldehyde (~2 Å) corroborates that MJ1647 forms tetramers (Fig. 1h).

Our data indicate that MJ1647 forms homotetramers in solution and that MJ1647 tetramerization depends on the C-terminal extension, which is henceforth referred to as tetramerization module TM. In addition, we show that MJ1647 cannot efficiently cross-oligomerise with canonical histones like A3.

**A3 and MJ1647 interact with 30- and 60-bp DNA fragments in vitro, respectively.** MJ1647 is highly divergent from canonical archaeal histones, including a substitution of residue R19 (HMfB numbering) of the so-called 'RT pair'. This motif is highly conserved in canonical histones and mediates DNA binding, which suggests that the DNA-binding mode of MJ1647 is different or that the DNA affinity is reduced[7,26,29,30] (Fig. 1a). We carried out Electrophoretic Mobility Shift Assays (EMSAs) with A3 and MJ1647 to compare their ability to form protein–DNA complexes with 30 bp and 60 bp double-stranded DNA fragments. The probes consist of native *M. jannaschii* genome sequences from the Rpo5 gene in the RNAP subunit operon. The canonical A3 histone formed a single-shifted species with a 30 bp DNA fragment, but two species with a 60 bp DNA fragment (Fig. 2a and Supplementary Fig. 3). This reflects one or two A3 dimers interacting with the probes where each dimer interacts with 30 bp of DNA, a binding pattern that is typical for

canonical archaeal histones. In comparison, MJ1647 formed only a smear with a 30 bp probe and a single species with the 60 bp probe. This indicates that the MJ1647-30 bp DNA complex is unstable and suggests that MJ1647 requires a longer DNA probe with at least 60 bp of DNA (Fig. 2b and Supplementary Fig. 3). Incubation of MJ1647ΔTM with 30 and 60 bp EMSA templates resulted in a binding pattern identical to A3, demonstrating that the MJ1647 core in the absence of the tetramerization module, MJ1647 has retained the DNA-binding mechanism of canonical histones (such as A3) interacting with 30 bp (Fig. 2c and Supplementary Fig. 2).

To analyse the footprint of A3 and MJ1647 on DNA further, we reconstituted archaeal chromatin using recombinant proteins and a 0.5-kb DNA fragment in vitro, and subjected it to MNase digestion (Fig. 2d). Binding of the A3 histone yielded a protection pattern with ~30 bp increments (~60, ~90, ~120 bp bands as prevalent species) in good agreement with canonical hypernucleosomes and similar to the native *M. jannaschii* chromatin digestion pattern (see below). In contrast, MJ1647 generated a ~60 bp increment pattern (~70 and ~130 bp) (Fig. 2e). The MJ1647ΔTM mutant yielded very weak fragmentation pattern related to WT MJ1647, but with higher MNase concentrations a smaller fragment appeared of ~35 bp corresponding to the protection site of a single histone dimer (Fig. 2f). The larger fragments obtained for MJ1647ΔTM are suggestive of interactions between flanking dimers in absence of the TM driving tetramerization. These results do not only confirm a 60 bp DNA-binding site of MJ1647, but also suggest that MJ1647 can establish higher-order filaments on DNA, possibly forming an unusual type of hypernucleosome. We tested direct interactions between MJ1647 tetramers during chromatinization by assembling MJ1647 chromatin on the 0.5 kb DNA fragments and subsequently incubating these complexes in presence of BS$^3$. MJ1647 cross-linked predominantly as tetramers, but in the presence of DNA, we observed higher-order crosslinks that are consistent with the formation of short MJ1647 filaments on DNA (Fig. 1i).

To test whether MJ1647 chromatinisation results in topological changes of the chromatinised DNA, we carried out EMSAs with larger DNA templates (linearised 5.5 kb plasmid DNA) and separated the complexes by agarose gel electrophoresis. The addition of either histone variant, A3 and MJ1647, led to faster migration of the DNA reflecting DNA compaction or possibly supercoiling (Fig. 2g, h). While increasing A3 concentrations resulted in gradually faster DNA migration, increasing MJ1647 concentrations resulted in an abrupt shift in DNA mobility between 8 and 12 µM MJ1647 indicating highly cooperative binding consistent with previous findings[26]. For MJ1647ΔTM we observed a more gradual response in DNA migration similar to A3 (Fig. 2i).

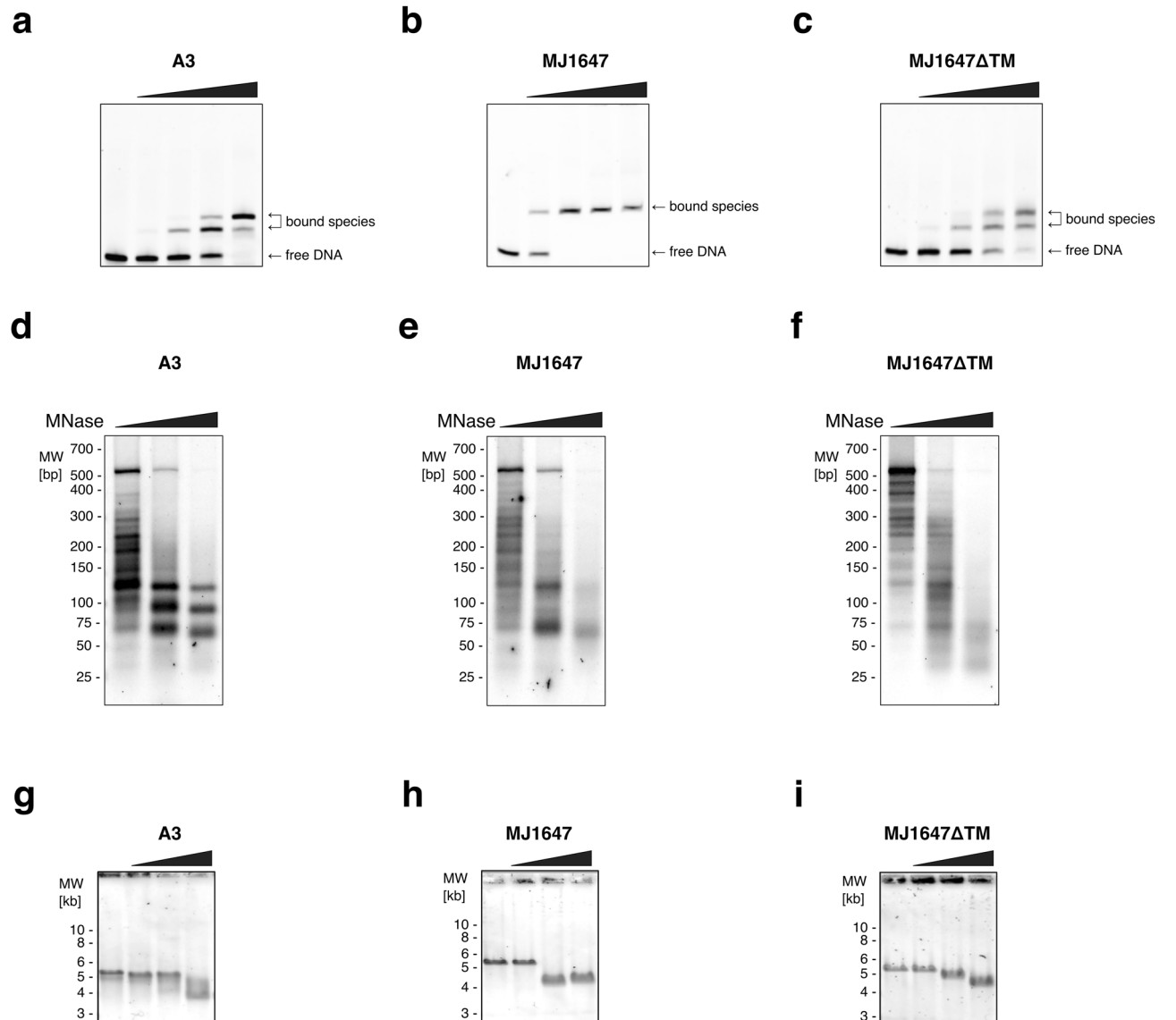

**Fig. 2 MJ1647 forms unusual 'hypernucleosome' polymers in 60 bp steps. a–c** EMSA experiments testing binding to 60 bp DNA templates. Increasing concentrations of histone A3 (2, 4, and 10 μM monomer) (**a**), MJ1647 (4, 8, 12 and 20 μM) (**b**) and the deletion variant MJ1647ΔTM (2, 4, 6 and 10 μM) (**c**) were incubated with the respective [32]P-labelled DNA templates and resolved on native polyacrylamide gels. **d–f** MNase digestion reveals a 60 bp protection pattern for MJ1647-chromatinised DNA. 1 μg of a ~500 bp PCR product was incubated with equistoichiometric amounts (histone dimer per 30 bp DNA) for histones A3 (**d**), MJ1647 (**e**) and MJ1647ΔTM (**f**) before exposure to increasing amount of MNase (0.1, 0.3, 1, 3 u). DNA from the digested chromatin was purified and resolved on agarose gels. **g–i** EMSAs with 5500 bp DNA templates testing DNA compaction. Increasing amounts of histones A3 (**g**), MJ1647 (**h**) and MJ1647ΔTM (**i**) were incubated with DNA and samples were resolved on agarose gels. Protein concentrations were 2, 4, 6 μM histone monomers for A3 and MJ1647ΔTM, 4, 8, 12 μM for MJ1647. DNA was visualised by post-staining with SYBR Gold. Representative gels of at least three replicates are shown.

**Single-molecule experiments reveal DNA compaction.** To test the DNA compaction properties of A3 and MJ1647, we carried out Tethered Particle Motion (TPM) experiments. This single-molecule technique reports on the length and conformation of dsDNA molecules tethered to a surface at one end by reporting the Root Mean Square motion (RMS) of a bead attached to the other end where the RMS is reduced upon DNA compaction[31]. First, we investigated the effect of binding of the A3 histone on DNA conformation. Experiments with A3 demonstrate a gradual progressive DNA compaction that is evident from a reduction in RMS upon titration of A3 and indicative of hypernucleosome formation (Fig. 3a). The RMS value under saturated conditions is ~80 nm, a value that is comparable to the one obtained with HMfB for the same DNA template[10]. The hypernucleosome

formation occurs with lower cooperativity compared to HMfB, which can be attributed to A3 having only one predicted stacking interaction (E35-K66) instead of three in HMfB (D14-R48, K30-E61 and E34-R65). For comparison, the residues mediating dimer–dimer interactions are conserved between HMfB and A3 (L47, H50, D60, L63 in A3). Titration of MJ1647 likewise leads to increasing DNA compaction, but strikingly we observed distinct populations corresponding to distinct successive binding events of MJ1647 (Fig. 3b). Overall, MJ1647 compacted DNA in a cooperative manner, although the distinct populations that we observed make it difficult to quantify this effect. To extract quantitative structural information on MJ1647 binding from these data we transformed the RMS into end-to-end distance (see ref. [10] and "Methods" and Fig. 3d, e). Importantly, the pairwise

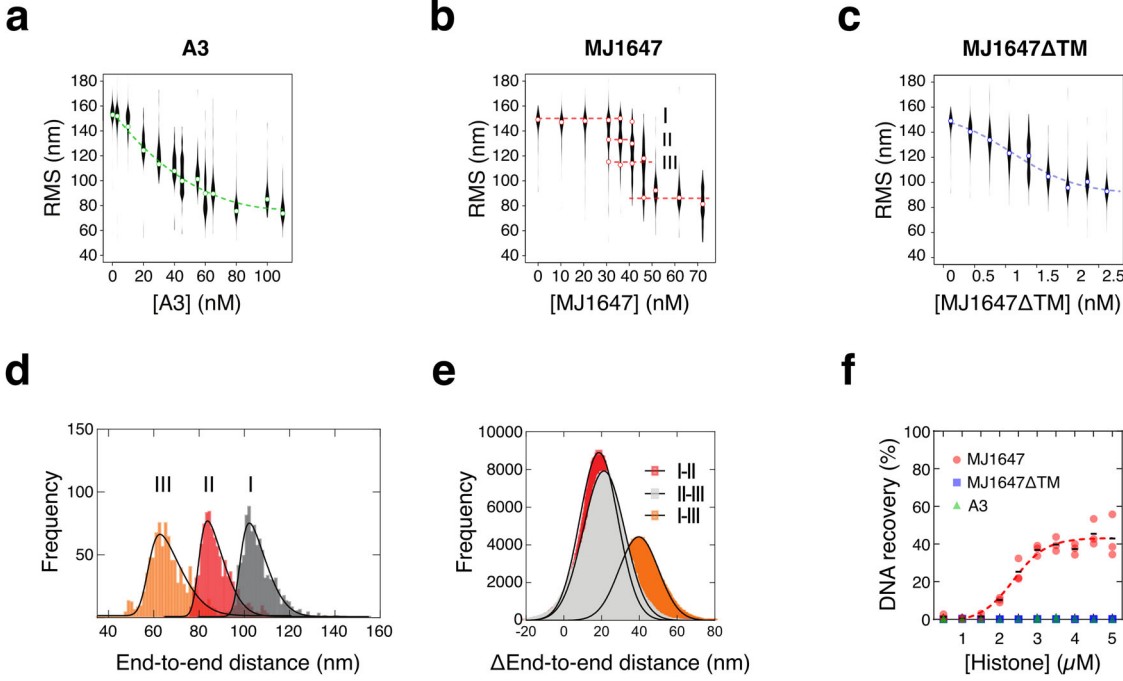

**Fig. 3 MJ1647 tetramers compact and bridge DNA duplexes. a–c** Tethered Particle Motion (TPM) experiments testing DNA compaction by A3 (**a**), MJ1647 (**b**) and MJ1647ΔTM (**c**). Root Mean Square displacement (RMS) values are plotted as a function of histone monomer concentration. The RMS values were obtained from fitting the data with a Gaussian distribution. Violin plots show the distribution of RMS measurements from individual beads with open circles showing the mean value. For MJ1647, mean values represent the mean for each of three distinct populations labelled with roman numbers (I, II and III). These three populations were used for end-to-end distance calculations (see panel **d**). Dashed lines are a line to guide the eye. **d** Histograms of calculated end-to-end distances of the three populations at ~150, 132 and 112 nm observed at 36 nM MJ1647 monomer. Of each population, the 25 beads closest to the fitted RMS value were selected and the end-to-end distance was calculated of the 2.5% most distant positions with respect to the centre of the beads. Histograms were fitted with a skewed normal distribution. **e** Pairwise distribution plot of the differences between the end-to-end distance peaks I, II and III from panel d. Histograms were fitted with a Gaussian distribution. **f** DNA-bridging assays. DNA recovery (%) is plotted as a function of histone monomer concentration. Black bars show mean values of three independent measurements for MJ1647 with coloured symbols showing the three individual replicates. Dashed line was included to guide the eye.

distance between the peaks is ~20 nm. Previous TPM experiments with tetramers of HMfB wrapping 60 bp showed an RMS reduction of ~20 nm[32,33]. Our data would thus be compatible with MJ1647 wrapping ~60 bp of DNA in a similar manner to canonical histones. A TPM experiment with MJ647ΔTM showed a smooth DNA compaction without visible steps, confirming that the deletion of the TM leads to a DNA-binding mode similar to that of canonical histone A3 and HMfB (Fig. 3c).

**MJ1647 but not A3 mediates DNA-bridging.** As MJ1647 forms tetramers in solution and interacts with 60 bp of DNA, we speculated whether MJ1647 could interact with DNA in trans by bridging two 30 bp DNA sites in addition, or alternatively, to binding to two adjacent 30 bp DNA-binding sites (i.e., 60 bp) in cis. We tested the bridging properties of histones using a radiolabeled DNA pulldown assay where a biotin-labelled DNA molecule is immobilised on magnetic streptavidin-coated beads[34–36]. Histone variants are added together with a second [32]P-labelled DNA, and if bridging between the two DNA species occurs, the [32]P-labelled DNA is immobilised on the beads and quantified as % recovery. The results show that MJ1647 is capable of bridging two DNA molecules in a fashion that is dependent on the TM, as neither A3 nor MJ1647ΔTM are capable of bridging DNA (Fig. 3f). Negative controls with MJ1647, but without biotin-labelled DNA, did not recover significant levels of [32]P-labelled DNA. The MJ1647 monomer concentration with a half-maximal recovery was ~2.5 µM which is higher than the concentrations required for DNA compaction in TPM assays

under single-molecule conditions. In part, this can be explained by the higher DNA concentrations used in the bridging assay that require a higher MJ1647 concentration to achieve saturation. Moreover, the general effect has been observed previously for other chromatin proteins known to bridge DNA such as H-NS, Rok, and MvaT[34,35,37].

**Structural basis of MJ1647 dimer and tetramer formation.** To elucidate the role of the MJ1647 tetramerisation module in oligomerisation and DNA binding, we solved the molecular structure of the MJ1647 histone by crystallisation and X-ray diffraction at 1.9 Å resolution (Fig. 4a, b and Table 1). MJ1647 crystallised as dimers and the refined model shows that each monomer contains the classic histone fold (three α-helices separated by two loops) with a very strong correspondence to well-characterised archaeal histones such as HMfB, including the electrostatic surface charge distribution (Fig. 4a, b). Unlike the disordered tails of eukaryotic histones, the MJ1647 tetramerisation module forms a well-structured region of the protein and is composed of two α-helices separated by a loop (Fig. 4a). The two helices from each monomer are packed against each other to create an interleaved or 'handshake' arrangement, adding considerably to the overall interaction surface between monomers in the dimer. The dimers in the crystal structure differ to the tetrameric state of the protein in solution observed in SEC-MALS and native mass spectrometry experiments (Fig. 1c, f). This discrepancy is possibly due to the low pH (4.6) in the crystallisation buffer that could favour MJ1647 dimers.

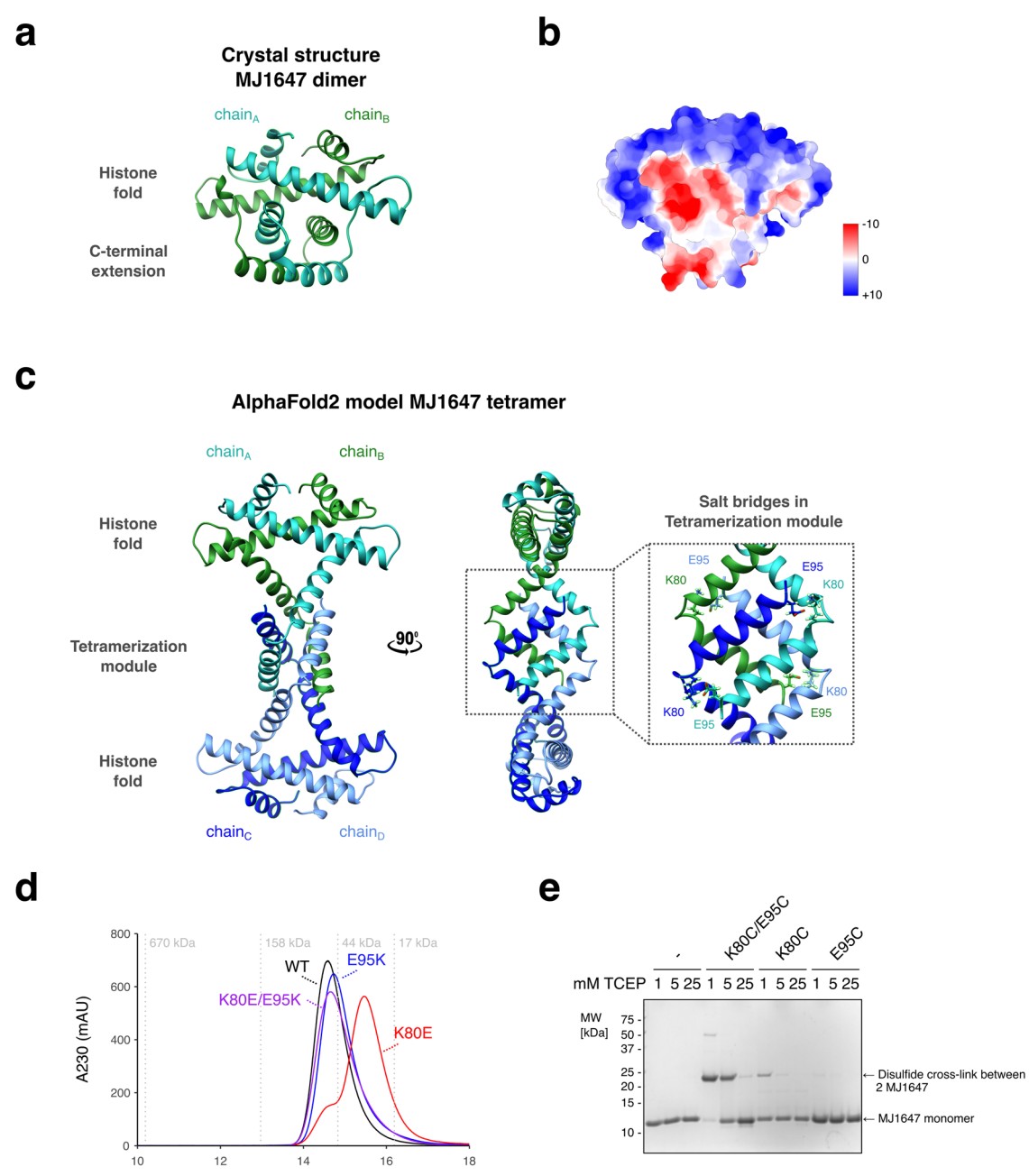

**Fig. 4 MJ1647 can form dimers and tetramers. a** Side view of the MJ1647 dimer crystal structure in ribbon representation with the DNA-binding interface on top. The two monomers are shown in sea green and dark green, respectively. **b** Side view of the electrostatic surface charge distribution of the MJ1647 dimer in (ranging from −10 (red) to +10 (blue) kcal/mol*e). **c** AlphaFold2 model of the tetrameric 'dimer of dimer' MJ1647 species with the two dimers (chain A/B and chain C/D) highlighted in dark green/sea green and blue/light blue, respectively. The tetramer is organised into two outward-facing histone folds and a central tetramerization module (TM). The zoom on the right shows the four salt bridges contributing to tetramer stability, these are formed between residues K80 and E95 between chains A and D, and chains B and C, respectively, in the predicted tetramer. **d** Size-exclusion chromatography elution profiles of wild-type MJ1647 and charge-reversal substitution variants of residues K80 and E95. The wild-type protein elutes as one sharp peak at the expected size for a tetramer of ~44 kDa. The E95K mutant forms broader peaks and elutes with longer retention times compared to wild type indicating destabilisation of the tetramer (purple and light blue trace, respectively). In contrast, the K80E/E95K double charge-reversal mutant (red) elutes identical to the wild-type protein (in green) by reestablishing the salt bridge and tetramer integrity. **e** Replacement of the predicted K80-E95 salt bridge with a disulfide bridge. Two cysteine residues present in MJ1647 were replaced by serine (C28S/C62S) and K80C, E95C or the K80C/E95C mutations were subsequently introduced. The MJ1647 cysteine mutants were incubated with 1, 5 or 25 mM TCEP reducing agent to distinguish labile non-structural versus structural disulfide bonds. The C28S/C62S variant was included as control (-). Proteins were subsequently analysed by non-reducing SDS-PAGE with Coomassie staining.

**Table 1 Data collection and refinement statistics (molecular replacement).**

| | MJ1647 dimer |
|---|---|
| *Data collection* | |
| Space group | P2$_1$ |
| Cell dimensions | |
| *a*, *b*, *c* (Å) | 42.0, 81.1, 53.5 |
| α, β, γ (°) | 90.0, 98.7, 90.0 |
| Resolution (Å) | 81.2–1.88 (1.91–1.88)[a] |
| $R_{sym}$ or $R_{merge}$ | 8.3% (104.2%) |
| *I* / σ*I* | 7.3 (1.2) |
| Completeness (%) | 100 (99.7) |
| Redundancy | 3.3 (3.3)[a] |
| *Refinement* | |
| Resolution (Å) | 53.0–1.88 |
| No. of reflections | 29,372 |
| $R_{work}$/$R_{free}$ | 21%/25.3% |
| No. of atoms | |
| Protein | 3116 |
| Ligand/ion | 0 |
| Water | 240 |
| *B*-factors | |
| Protein | 37.9 |
| Ligand/ion | N/A |
| Water | 39.4 |
| R.m.s. deviations | |
| Bond lengths (Å) | 0.006 |
| Bond angles (°) | 0.764 |

[a]Values in parentheses are for the highest-resolution shell.

Unable to obtain crystals of the MJ1647 tetramer at higher pH, we generated structural models for MJ1647 tetramers using the structure prediction algorithm AlphaFold2[38–41] (Fig. 4c and Supplementary Fig. 3). In the MJ1647 tetramer model, the two DNA-binding histone folds are located opposite each other and connected by the four MJ1647 TMs. While TM helices 4 and 5 form monomer–monomer interactions within MJ1647 dimers in the crystal structure, TM helices 4 and 5 'open up' in the tetrameric model and make new interactions with opposing monomers that enable dimer–dimer interactions within the tetramer (Fig. 4c). The predicted aligned error (PAE) indicates high confidence (PAEs <10 Å) in the relative positions of the residues in the tetramerization module (Supplementary Fig. 4). The interface between the two dimers consists of the tetramerization module's hydrophobic core that is stabilised by four salt bridges between residues K80 and E95, both of which are strictly conserved in all MJ1647 homologues (Fig. 4c). In addition, K68 makes polar contacts with the backbone of T93 and L96 of the tetramer model, albeit with low confidence for side chain orientation for residues 68–80.

To validate the tetrameric model of MJ1647 experimentally, we produced two charge-reversal mutants (K80E and E95K) and the K80E/E95K double mutant and assessed their oligomeric state by SEC (Fig. 4d). While the wild-type MJ1647 elutes in one sharp peak corresponding to the expected molecular weight for a tetramer of 44 kDa in our SEC-MALS experiments, the K80E substitution resulted in a delayed and broader elution profile suggestive of weakened tetramerization. In comparison, reinstating the salt bridge by introducing a double charge-reversal K80E/E95K lead to a sharp peak with a retention time corresponding to that of the tetramer. The E95K single charge-reversal mutant, however, was not significantly affected.

We tested the DNA-binding properties of the salt bridge mutants using EMSA and DNA-bridging assays. At lower protein concentrations both the K80E and E95K charge-reversal substitutions formed a DNA-protein complex in EMSAs that migrated faster than the tetramer-DNA complex and that likely corresponds to DNA-bound dimers (Supplementary Fig. 6). With higher protein concentrations, both mutants formed a second complex corresponding to the tetramer-DNA complex indicating that dimer–dimer interactions are weakened but not fully disrupted. All three mutants retained the ability to bridge DNA, although with different affinities (Supplementary Fig. 7). Our finding that the salt bridge mutants retain the ability to bridge DNA further suggest that the salt bridge mutations alone do not disrupt tetramerization entirely and a lower affinity to form tetramers might be still sufficient in the context of the larger complexes formed between multiple MJ1647 and two dsDNA molecules.

To provide additional evidence for tetramerization according to the AlphaFold2 model, we probed the proximity of K80 and E95 by introducing cysteine pairs at positions 80 and 95. We first removed the two cysteine residues present in MJ1647 (C28S/C62S) and subsequently introduced K80C and E95C substitutions. All mutant variants were expressed at high levels, soluble and thermostable, and eluted in SEC similar to wild-type MJ1647 consistent with a tetrameric state. The K80C mutation caused a slight destabilisation of the tetramers similar to K80E while the K80C/E95C double mutation was monodisperse and eluted similar to the C28S/C62S (Supplementary Fig. 5). The predicted C-alpha distance between residue 80 and 95 in the AlphaFold2 tetramer model (7.3 to 7.7 Å) is within the range of disulfide bonds of 3.5–7.5 Å[42] (Fig. 4c). By comparison, the distance between positions 80 and 95 of monomers within the dimer of the crystal structure (11.1 Å) is incompatible with disulfide bond formation. Figure 4e shows the SDS-PAGE analyses of the double-cysteine substitution at three different reducing agent concentrations (1, 5 and 25 mM TCEP), compared to the two single-cysteine substitutions and the C28S/C62S background as negative controls. The SDS-PAGE reveals the double-cysteine variant as covalently linked dimeric MJ1647 species, a result that according to the distance information discussed above is only compatible with the predicted model of the tetramer and which runs as a dimer under the denaturating conditions of SDS-PAGE (Fig. 4e). This covalent dimer is very stable and resistant to 1 and 5 mM TCEP, and it requires high concentrations (25 mM) to reduce the disulfide bond to generate monomeric MJ1647. One of the single-cysteine negative control substitutions, K80C, shows a very weak covalent dimer band at low (1 mM) TCEP which is effectively reduced at intermediate (5 mM) TCEP concentrations, indicating a non-specific disulfide bond formation between surface exposed cysteine residues between non-associated proteins in solution.

In summary, multiple analyses prove that MJ1647 is tetrameric in solution, including SEC-MALS, native mass spectrometry, BS$^3$ cross-linking, computational modelling, and mutagenesis analysis of salt bridges such as charge-reversal interference and cysteine disulfide bridge variants.

**M. jannaschii chromatin is dominated by dynamic hypernucleosomes.** Canonical histones appear to be the dominant chromatin proteins in *M. jannaschii*. To test whether potential MJ1647-chromatinised regions can be identified in the chromatin landscape of *M. jannaschii*, we carried out MNase-seq experiments by digesting chromatin isolated from *M. jannaschii* cells with micrococcal nuclease (MNase) and deep-sequencing the MNase-resistant DNA fragments over a wide size range up to 600 bp in paired-end mode as described previously[43]. For technical reasons, DNA fragments below 100 bp were depleted in the samples. We tested different MNase concentrations (3 and 8 units/ml) with nucleoids prepared from *M. jannaschii* in logarithmic growth phase yielding

**a**

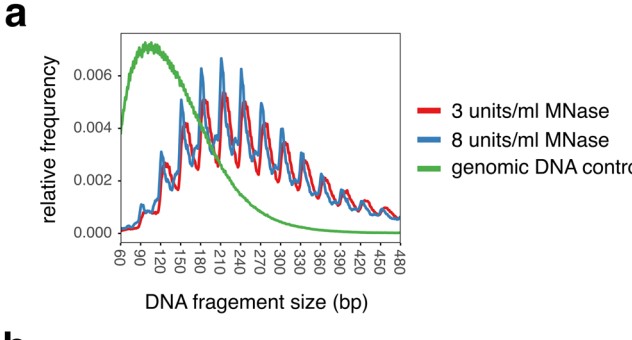

**b**

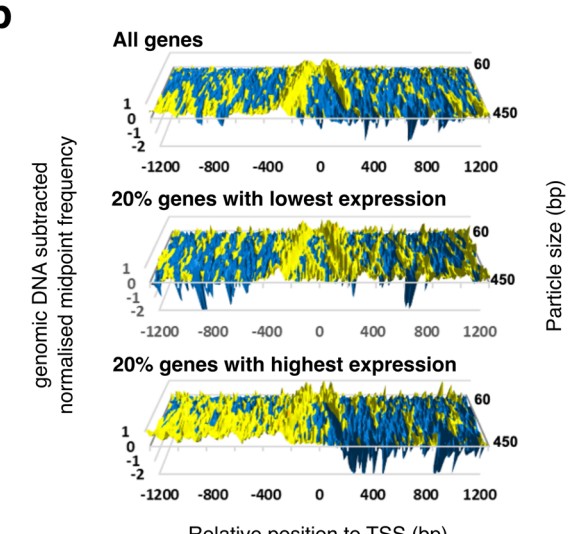

**Fig. 5 The chromatin landscape of *M. jannaschii*. a** MNase-resistant DNA species from *M. jannaschii* chromatin span a wide size range and exhibit 30 bp periodicity consistent with variably-sized hypernucleosome organisation. MNase-seq fragment size versus relative frequency is plotted for deproteinised genomic DNA and bulk chromatin from cells in log growth phase treated with 3 or 8 units/ml MNase. **b** Chromatin particles are positioned with respect to transcription start sites and level of transcription. Normalised chromatin particle mid-point frequencies (y axis) from logarithmic growth phase cells were plotted relative to transcription start sites (10 bp bin size) across increasing chromatin particle sizes (15 bp bins). To ameliorate MNase cleavage bias, the signal for deproteinised genomic DNA was subtracted[72]. Values >0 are coloured yellow. Transcript level estimates were obtained from previously published RNA-seq data[28].

essentially superimposable profiles of DNA fragment size distribution with peaks for sizes differing by multiples of 30 bp (Fig. 5a). No enrichment of 60 bp multiples (as expected for extensive MJ1647 chromatinisation) was detectable in the global distribution and the overall pattern was similar to MNase-seq experiments for *T. kodakarensis*[11] that possesses canonical histones but no MJ1647 homologue. To test whether specific genomic regions show enrichment of 60 bp multiples, we calculated the log$_2$-fold ratio of DNA fragment coverage for 60 bp multiples (120 and 180 ± 5 bp) compared to odd 30 bp multiples (90, 150, and 210 ± 5 bp) over 120 bp bins. The data correlated well between the samples treated with different MNase concentrations (Pearson's rho = 0.75, Supplementary Fig. 8). The ratio between 60mer and odd 30mer coverage showed a tight distribution in the ratio (interdecile range of 2.61 and 2.59 for the two samples, respectively). No distinct cluster of genomic bins with stronger bias to 60mer multiples serving as potential MJ1647 binding sites could be identified.

Although *M. jannaschii* chromatin particles are not organised into regular arrays with respect to each other, they are organised

with respect to promoter regions. MNase-seq chromatin particle frequency surrounding transcriptional start sites (TSSs)[28] revealed that promoter regions are generally flanked on one or both sides by MNase-resistant chromatin particles of various sizes while the TSS itself is chromatin free as in other archaea[2,11,44]. Importantly, this positional bias appeared to be dependent on the transcription level. The top 20% transcribed genes including the rRNA operons are generally more depleted of hypernucleosomes in the region downstream of the TSS compared to the bottom 20% of transcribed genes (Fig. 5b and Supplementary Fig. 9).

**MJ1647 and A3 inhibit transcription by impairing promoter access.** MNase-seq data shows that histone occupancy anticorrelates with mRNA levels in *M. jannaschii* (Fig. 5b) consistent with a putative regulatory function[16,18,22,23]. From first principles, histones can compete with transcription pre-initiation complexes (PIC) for access to the promoter and they might also present a barrier for transcription elongation complexes. In bacteria, chromatin (or 'nucleoid') proteins such as *E. coli* H-NS occlude promoters but can also effectively stall transcription elongation complexes, possibly by trapping them in topologically closed domains formed through DNA-bridging[45].

In order to elucidate the effect of *M. jannaschii* histones on transcription, we performed multi-round in vitro transcription assays using linearised plasmids with the SSV T6 promoter fused to a 500 bp transcribed region derived from the *M. jannaschii* RNAP subunit rpo2 gene that generate a run-off transcript of 574 nt. At 1:1 stoichiometric ratio between histone dimers and hypothetical 30 bp DNA-binding sites, the transcript yield was reduced by >50% for all three histones (A3, MJ1647 and MJ1647ΔTM) while a two-fold excess of histones lead to almost total inhibition of transcription (Fig. 6a). To test whether this inhibitory effect stems from impaired transcription elongation or competition for promoter access, we conducted synchronised single-round transcription assays that can detect reduced transcription elongation as well as EMSAs testing PIC assembly in the presence of histones.

Synchronised single-round in vitro transcription assays were carried out essentially under the same conditions as multi-round assays but included a pre-incubation step with only limiting nucleotides being present (100 μM ATP and GTP) allowing for the PIC to progress and synthesise a 6nt initial transcript. These initially transcribing PICs were allowed to form in the presence of histones in 1:1 stoichiometric ratio between histone dimers and the predicted number of 30 bp DNA-binding sites (Fig. 6b). Simultaneously with the addition of the full nucleotide set that allows transcription to resume, reinitiation was blocked by the addition of an excess of a TFB variant lacking the N-terminal zinc-ribbon domain that binds the DNA-TBP complex but blocks RNAP recruitment[46]. This assay allowed us to track transcription elongation through the accumulation of full-length run-off transcripts over time. In sharp contrast to the effects that we observed in multi-round transcription experiments, the accumulation of full-length transcripts was only very marginally affected by histones. Likewise, the presence of histones did not lead to the appearance of smaller transcripts of specific sizes suggesting that pausing or premature termination are largely absent under these conditions. We concluded that the inhibitory effect of histone chromatinization on *M. jannaschii* transcription in vitro is largely due to interference with transcription initiation.

**Discussion**
In contrast to eukaryotes that utilise the ubiquitous histone octamer as the fundamental chromatin building block, archaea employ a range of different chromatin proteins to enable genome

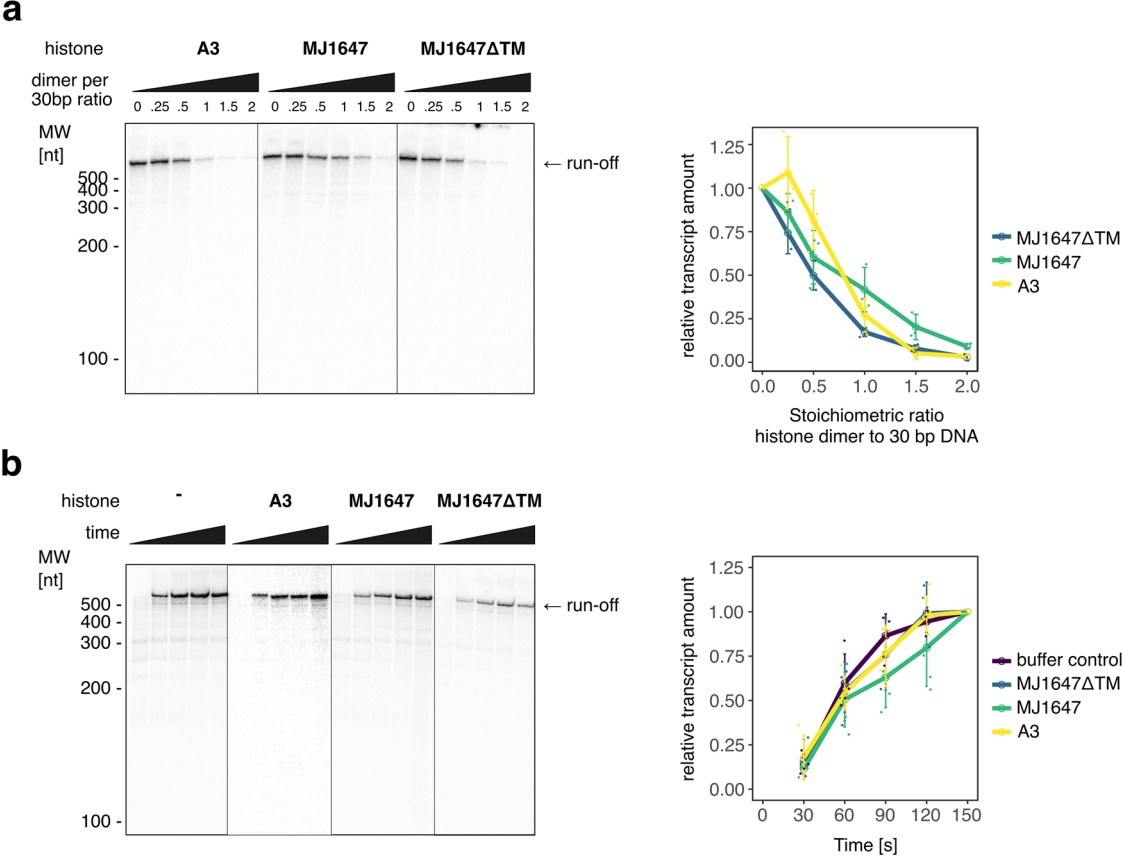

**Fig. 6 A3 and MJ1647 compete with the transcription machinery for promoter access. a** Multi-round in vitro transcription assays testing transcription inhibition by A3, MJ1647 and MJ1647ΔTM. Transcription assays contained RNA polymerase, initiation factors TBP, TFB and TFE and a plasmid DNA template harbouring the strong T6 promoter fused to a ~500 bp sequence derived from *M. jannaschii*. Histones were added at the indicated stoichiometric ratios of histone dimer per 30 bp DNA. Transcripts were purified and resolved on a 7 M Urea, 1× TBE sequencing gel. A representative gel of three biological replicates is shown along the quantification of transcripts normalised to the transcript level in absence of histones. **b** Synchronised single-round in vitro transcription assays testing the effect of A3, MJ1647, and MJ1647ΔTM chromatinization on transcription elongation. Samples were taken in 30 s increments from 30 s to 150 s and resolved on a 7 M Urea, 1× TBE sequencing gel. A representative gel of at least four biological replicates per histone is shown along the quantification of transcripts normalised to the signal at the 150 s time point.

compaction and gene regulation[1]. In most archaeal phyla this includes histones, often a combination of well-characterised canonical histones and variants with largely unknown structure and function. Here we have set out to systematically compare the canonical A3 histone from *M. jannaschii* and the MJ1647 variant that piqued our interest due to its unusual C-terminal extension.

We solved the X-ray structure of MJ1647 dimers at 1.9 Å resolution, which reveals the C-terminal extension to be well-folded forming two short helices that stabilise monomer–monomer interactions within the dimer with a handshake motif (Fig. 4). In contrast to in crystallo, MJ1647 forms tetramers in solution and this is dependent on the C-terminal extension which we refer to as tetramerization module (TM). In our high-confidence model of tetrameric MJ1647 generated by AlphaFold2, two MJ1647 dimers interact via the TM, where the two TM helices hinge open and form contacts with the TM helices of the opposite dimer. This arrangement is stabilised by four salt bridges. We confirmed the AlphaFold2 model by introducing charge-reversal and double charge-reversal substitutions that impaired and restored tetramers, respectively, as well as engineered disulfide bridges replacing the salt bridges (Fig. 4). Overall, our crystallisation results suggest that MJ1647 might exist in an alternative state next to the dominant tetrameric state we observed in our biochemical assays. Regulating the equilibrium between tetrameric and dimeric states could be a way to regulate the DNA-bridging activity of MJ1647 in the cell.

We compared the DNA-binding properties of A3 and MJ1647 using EMSAs and by MNase digestion of in vitro reconstituted chromatin. While A3 dimers interact with and protect 30 bp of DNA, MJ1647 tetramers interact with and protect 60 bp of DNA (Fig. 2). At the single-molecule level, TPM results demonstrate that A3 compacts longer DNA fragments in a cooperative fashion and in seemingly small increments. In contrast, MJ1647 shortens the DNA in experimentally discernible larger steps consistent with 60 bp DNA wrapping. This more extensive DNA wrapping behaviour is critically reliant on the TM (Fig. 3). The tetrameric AlphaFold2 model of MJ1647 shown in Fig. 4 places the histone folds of the two dimers at diametrically opposing ends of the molecule, in agreement with the DNA-bridging property of MJ1647. This model cannot fully account for the 60 bp protection of DNA if the MJ1647 histone fold were to interact with the DNA in the same mode as canonical histones like A3. We surmise that MJ1647 tetramers use an alternative DNA-binding mode compared to the canonical histone-DNA interactions observed in A3 and the dimeric MJ1647ΔTM mutant (Fig. 7). Alternative DNA-interaction modes among divergent paralogues of DNA-binding proteins are not unknown, and a highly unusual DNA-binding mode for a histone has indeed recently been reported in the bacteria *Bdellovibrio bacteriovorus* and *Leptospira interrogans*[4]. Furthermore, the 60 bp protection pattern could reflect other, possibly higher-order oligomers made of MJ1647 tetramers occurring in the

## DNA bridging

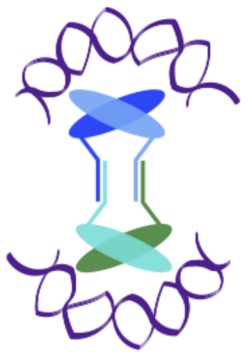

## DNA compaction

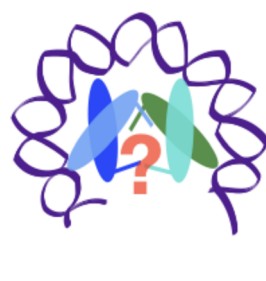

**Fig. 7 A model for MJ1647-DNA-bridging and compaction by alternative DNA-binding modes.** MJ1647 bridges DNA by positioning the histone folds opposite each other with the TM adopting a conformation as predicted in the AlphaFold2 model (Fig. 4c). DNA compaction as observed in the TPM experiments (Fig. 3b) likely occurs via canonical histone dimer–dimer interactions. Such dimer–dimer interactions would require a rearrangement or breaking of the TM due to steric clashes (as indicated by the red question mark).

presence of long DNA fragments used in the chromatin reconstitution and TPM experiments (Fig. 2e and Fig. 3). We obtained experimental evidence for the formation of higher-order oligomers in a strictly DNA-dependent manner in BS[3] cross-linking experiments (Fig. 1i). DNA-bridging assays unequivocally demonstrate that MJ1647, but not A3, is able to bridge two DNA duplexes which likely enables the formation of DNA loops in vivo (Fig. 3). This makes MJ1647 to the best of our knowledge the first histone able to form inter-doublestrand DNA connections. A recent bioinformatics study of novel histone variants revealed several other candidate histones that could potentially tetramerise via C-terminal extensions[47].

MJ1647 does not heterodimerise with A3 suggesting that *M. jannaschii* partitions its chromatin into canonical histone and MJ1647 filaments similar to the partition between histones and the NAP TrmBL2 found in *T. kodakarensis*[48]. MJ1647-DNA complexes form in a highly cooperative manner (Fig. 3b) and cooperativity appears to depend on the TM. Potential MJ1647 filaments must be different from canonical hypernucleosomes as the TM would lead to steric clashes at the hypernucleosome core. The 60 bp DNA binding and compaction by MJ1647 tetramers as the base unit of these filaments is nevertheless reminiscent of tetramers formed by canonical histones.

What drove the evolution of MJ1647 homologues in Methanococcales? Growth temperature seems to be a major driving force for the evolution of additional NAPs and histones in archaea[49]. In line with this idea, the ancestor of Methanococcales, the lineage where MJ1647 evolved, is predicted to have been a thermophile[50]. Notably, MJ1647 homologues were retained not only in thermophilic Methanococcales species, but also those Methanococcales species adapting to mesophilic growth conditions such as *Methanococcus maripaludis*. Archaea lack linker histones and hypernucleosomes do not self-associate unlike eukaryotic nucleosomes[9]. Our finding that MJ1647 tetrameric histones mediate DNA-bridging opens the possibility that MJ1647 plays a role in higher-order organisation of archaeal chromatin. The best characterised archaeal NAP mediating DNA-bridging is Alba[51–54]. Alba constitutes a large fraction of chromatin in many archaeal species such as *Sulfolobus shibatae* where it constitutes 4% of all cellular protein[55]. In contrast, shotgun

proteomics and quantitative immunodetection data suggest much lower expression levels for Alba in *M. jannaschii*[27,56]. Expression levels of Alba in the mesophile *M. maripaludis* are even lower at about 0.01% of total cellular protein coinciding with Alba evolving to acquire DNA sequence specificity in binding[56]. It is tempting to speculate that MJ1647 homologues functionally replaced Alba in its role of higher-order DNA compaction.

Our results show that A3 forms hypernucleosomes in vitro despite providing fewer stacking interactions than other histones including HMfB (Fig. 2). This is corroborated by our nucleosome sequencing data that reveal extended hypernucleosome formation in vivo (Fig. 5). In agreement with other euryarchaea such as *T. kodakarensis*[11,12], the MNase protection anticorrelated with mRNA levels, and highly active transcription units including rRNA operons were largely devoid of chromatin (Fig. 5b and Supplementary Fig. 9). Importantly, the gene promoters were nucleosome-free regions.

In *T. kodakarensis*, hypernucleosome chromatinization can inhibit transcription elongation in vitro and this inhibition is counteracted by transcription elongation factors Spt4/5 and TFS[16]. In contrast, we observed only a mild retardation of RNA polymerase by DNA chromatinization by both the canonical histone A3 and MJ1647. These differences might be explained by technical details that restrict the in vitro transcription experiments. In particular the choice of DNA templates containing high-affinity binding sites within the transcribed sequence used in previous experiments might explain these differences as hypernucleosomes formed on high-affinity binding sites show different DNA compaction properties compared to hypernucleosomes that form on non-specific DNA[33]. But it is tempting to speculate that the single stacking interaction within A3 might render A3 hypernucleosomes less impeding to transcription elongation. In *E. coli*, DNA-bridging NAP H-NS can effectively pause transcription elongation by forming bridged filaments[45]. This effect is concentration-dependent as higher H-NS concentrations favour the formation of linear, non-bridged filaments that provide less resistance to transcription elongation. In contrast, transcription inhibition by MJ1647 increases with protein concentrations (Fig. 6).

Rather than transcription elongation, we find that histone chromatinization in *M. jannaschii* attenuates transcription initiation, even for the very strong T6 promoter used in our assays. This is in line with a previous study showing that the canonical histones from *M. jannaschii* efficiently block access to the *rb2* promoter thereby increasing its dependency on transcription activator Ptr2[22]. The nucleosome-free regions around promoter regions in *M. jannaschii* might thus be the product of competition between histones and the transcription machinery for promoter access. In contrast, a comparison of MNase-seq data from isolated and in vitro reconstituted chromatin formed by HtkA and HTkB from *T. kodakarensis* found that the DNA sequence appears to be the main determinant for the formation of nucleosome-free regions[12]. Our MNase-seq data reveal a negative correlation between histone occupancy at promoters and expression levels (Fig. 5). While this relationship is not necessarily causal, it is consistent with histones regulating transcription in *M. jannaschii* via competition for promoter access. Recent ATAC-seq data for *H. volcanii* rather point to an accessible promoter state independent of the transcription level of the gene[57].

The call is still out whether histones in the archaea primarily function as genome compactors or gene expression regulators. The structure-function studies of histone variants, including histones from Asgård archaea, promise a rich hunting ground for discoveries including the origin and evolution of eukaryotic chromatin.

## Methods

**Chromatin isolation and MNase digestion.** *M. jannaschii* DSM 2661 cells were grown in 100 l fermenters in minimal medium containing 0.3 mM $K_2HPO_4$, 0.4 mM $KH_2PO_4$, 3.6 mM KCl, 0.4 M NaCl, 10 mM $NaHCO_3$, 2.5 mM $CaCl_2$, 38 mM $MgCl_2$, 22 mM $NH_4Cl$, 31 μM $Fe(NH_4)_2(SO_4)_2$, 1 mM $C_6H_9NO_6$, 1.2 μM $MgSO_4$, 0.4 mM $CuSO_4$, 0.3 μM $MnSO_4$, 36 nM $FeSO_4$, 36 nM $CoSO_4$, 3.5 nM $ZnSO_4$, 4 nM $KAl(SO_4)_2$, 16 nM $H_3BO_3$, 42 μM $Na_2SeO_4$, 0.3 nM $Na_2WO_4$, 11 μM $NaMoO_4$, 44 μM $(NH_4)_2Ni(SO_4)_2$ and 2 mM $Na_2S$ supplied with $H_2$:$CO_2$ gas in a 4:1 ratio at 85 °C[28]. In all, 0.2 g wet cells (wet weight) were resuspended in 5 ml PBS supplemented with EDTA-free protease inhibitor cocktail (Thermo Fisher) and centrifuged at $1500 \times g$ for 10 min at 4 °C to remove larger sulphide precipitates. These steps were repeated for a total of five washes. The final supernatant was then centrifuged at $14,000 \times g$ for 10 min at 4 °C. The cell pellet was resuspended in 1.25 ml extraction buffer (25 mM HEPES pH 7.0, 15 mM $MgCl_2$, 100 mM NaCl, 0.4 M sorbitol, 0.5% Triton X-100) supplemented with protease inhibitor and incubated at 4 °C for 30 min. Subsequently, 100 μl aliquots were centrifuged at $14,000 \times g$ for 15 min at 4 °C and the supernatant was removed. The chromatin pellets were resuspended in 50 μl extraction buffer with protease inhibitor and subjected to MNase digestion. For the naked genomic DNA control, DNA was extracted from the resuspended chromatin pellets using the DNeasy blood and tissue kit (Qiagen) according to the manufacturer's bacteria protocol.

In total, 50 μl resuspended chromatin or 6 μg genomic DNA was digested with the indicated amounts of MNase (Thermo Fisher) in 100 μl MNase buffer (20 mM Tris-HCl pH 7.4, 25 mM NaCl, 5 mM $CaCl_2$) at 37 °C for 30 min before the addition of 10 μl stop solution (2% SDS, 0.1 M EDTA, 0.1 M EGTA). DNA was purified using the QIAquick PCR purification kit (Qiagen) and eluted in 50 μl $H_2O$. The QIAquick PCR purification protocol selectively depletes dsDNA fragments <100 bp. The yield was assessed based on $A_{260}$ absorption and the fragment size distribution was assessed by resolving the samples on 3% agarose gels post-stained with ethidium bromide.

**MNase-seq library preparation and sequencing.** MNase-seq adaptor libraries were prepared using NEBNext reagents (New England Biolabs) according to the manufacturer's instructions, except that libraries were size selected on polyacrylamide gels to preserve the full-size distribution of MNase digested fragments. DNA was sequenced in 100 nucleotide paired-end mode on an Illumina HiSeq 2000 using v3 TruSeq SBS reagents by the University of Exeter, UK, DNA Sequencing Service.

Paired reads were trimmed to the 5' 25 bases and aligned to an index created from NCBI RefSeq chromosome (NC_000909.1) and extrachromosomal/plasmid element (NC_001732.1 and NC_001733.1) sequences using Bowtie v1.2.2[58] with command line flags: -v 2 --trim3 75 --maxins 1000 --fr -k 1 --sam. Aligned reads pairs numbers for log phase cells digested with 3 units/ml MNase, log phase cells digested with 8 u/ml MNase and deproteinised genomic DNA digested with MNase were: 30,923,719, 34,161,397 and 12,482,569, respectively. Aligned read pairs were sorted according to DNA element and into "chromatin particle size" classes based on paired-read end-to-end distance value +/− 10%. Frequency distributions of the mid-point positions between read pairs were calculated to define "chromatin particle positions" as described by, and using Perl scripts from ref. [43], and lightly smoothed by taking a 3-bin moving average.

Data were rendered as previously described by, and using Perl scripts from ref. [43]. Histograms (1 bp bins) of paired-read insert frequency versus paired-read insert size were plotted from equal numbers (10 million per dataset) of read pair alignments randomly selected with respect to map position to allow direct comparisons on the same graph $y$ axis. Particle position frequency distribution heatmaps were rendered with the Integrated Genome Browser[59] in Blue/Yellow mode with $y$ axis scale set by Percentile with Min = 0 and Max = 99. Genomic positions of chromatin particles specific to the 150 bp size class were defined as summit positions of peaks in paired-read mid-point frequency with top 1% of values. Cumulative particle frequency distributions were normalised by dividing by the average cumulative frequency value obtained for all bins surrounding the feature as described by Kent et al.[43]. to allow values from different size classes to be plotted on a common surface graph $y$ axis.

**Heterologous protein expression and purification.** The genes for A3 (MJ1258), MJ1647 and MJ1647ΔTM were cloned into pET-21 a(+) (Merck) via *Nde*I and *Xho*I restriction sites for the heterologous expression of untagged protein (Supplementary Table 1). Heterologous expression was carried out in *E. coli* Rosetta 2(DE3) and proteins were purified on a HiTrap Heparin affinity HP column (Cytiva)in N(200) buffer with a salt gradient to 1 M NaCl. The buffer of the eluted protein was exchanged to 20 mM MES/NaOH pH 6.5, 100 mM NaCl and proteins were purified by cation exchange chromatography on a HiTrap SP Sepharose column (Cytiva). Finally, proteins were purified by size-exclusion chromatography on a Superose12 10/300 column (Cytiva) with N(250) as running buffer. Plasmids for the expression of MJ1647 variants were generated using site-directed mutagenesis or NEBuilder HiFi DNA assembly (see Supplementary Table 1).

For the co-expression of MJ1647 and A3, the corresponding genes were cloned into pETDuet-1 (Merck) via *Nco*I and *Xho*I or *Nco*I and *Bam*HI restrictions sites, respectively. Co-expression and purification was carried as described for the single proteins. RNAP, TFB, TBP and TFE were purified as described previously[28,60].

**Breaking of disulfide bonds with TCEP.** In total, 6 μM protein was incubated with the indicated TCEP concentrations at room temperature for 30 min before the addition of 0.5 vol 3× non-reducing SDS-PAGE loading dye. Samples were incubated at 95 °C for 5 min before SDS-PAGE on 14% Tris-Tricine gels and subsequent Coomassie staining.

**SEC-MALS.** SEC-MALS analysis was carried out on a system equipped with an OPTILAB T-rEX differential refractometer and a DAWN-HELEOS 8+ laser photometer (Wyatt Technology. 100 μl of A3, MJ1647 or MJ1647ΔTM at 2.0 mg/ml was loaded onto a Superose12 10/300 column (Cytiva) in buffer N(250) at a flow rate of 0.5 ml/min. The differential refractive index was used to determine the protein concentration and the molecular weight was calculated using the ASTRA software v6.0.3 (Wyatt Technology).

**Native mass spectrometry experiments.** For native MS experiments, purified protein constructs were buffer exchanged using Bio-Spin P-6 columns (Bio-Rad) into 0.5 M ammonium acetate. Samples were analysed on the first-generation Synapt mass spectrometer (Waters). Samples were introduced into the mass spectrometer by direct injection method using in house prepared capillaries (borosilicate glass, 1.0 mm × 0.78 mm, Harvard apparatus) created using a needle-puller (P97, Sutter Instruments) and coated with gold using a sputter-coater (SC7620, Emitech) as described previously[61]. The Synapt instrument was externally calibrated using a 30 mg/mL solution of caesium iodide. Acquisition parameters were as following: capillary voltage 1.2 kV, cone voltage 40 V, extraction cone voltage 1 V, trap/transfer collision energy 6/4 V, bias voltage 4 V and source

temperature 40 °C. Mass spectra were analysed using MassLynx software v4.1 (Waters).

**EMSAs**. 20 µl samples contained the indicated histone concentrations and 50 nM Cy3-labelled dsDNA template in 10 mM Tris/HCl pH 8.0, 222 mM NaCl, 0.4 mM TCEP, 0.1 mg/ml BSA, 5% glycerol. Samples were incubated at 37 °C for 15 min. After the addition of 7 µl 4× native loading dye (20% Ficoll-400, 0.125 M Tris/HCl pH 6.8), samples were resolved on native Tris-glycine gels. For EMSAs with MJ1647 K80E and E95K (Supplementary Fig. 6) the 60 bp DNA templates were 5'-end radiolabelled with $^{32}$P instead of Cy3.

For EMSAs with larger DNA fragments, a 5500 bp region from the *M. jannaschii* genome was PCR-amplified with flanking *Nco*I and *Xho*I restriction sites and cloned into vector pGEM-T (Promega) (Supplementary Table 1). The template was excised from the vector via *Nco*I/*Xho*I restriction digest and isolated by agarose gel electrophoresis.

Overall, 20 µl samples contained the indicated histone concentrations and 40 ng of dsDNA template in 10 mM Tris/HCl pH 8.0, 110 mM NaCl, 0.4 mM TCEP, 0.1 mg/ml BSA. Samples were incubated at 37 °C for 15 min. After addition of 7 µl 4× native loading dye (20% Ficoll-400, 0.125 M Tris/HCl pH 6.8), samples were resolved on 0.8% agarose gels in 1× TAE buffer at 15 V for 16 h at room temperature. DNA was visualised using SYBR Gold stain (Thermo Fisher).

**DNA substrates for TPM**. Tethered particle motion and bridging assay experiments were performed using a 47% GC 685 bp DNA substrate described earlier[10]. The DNA substrate was generated by PCR using Thermo Scientific® Phusion® High-Fidelity DNA Polymerase and the products were purified using the GenElute PCR Clean-up kit (Sigma-Aldrich).

**TPM**. Measurements were performed as previously described[31,62] with minor modifications. Briefly, the flow cell was washed with 100 µL experimental buffer (50 mM Tris-HCl, pH 7 and 75 mM KCl) to remove excess beads and 100 µL protein diluted in experimental buffer was flowed in and incubated for 10 min. Next, the flow cell was washed with protein solution one more time, sealed with nail polish. After incubation, the flow cell was directly transferred to the holder and incubated for five more minutes in the instrument to stabilise the temperature at 25 °C for the measurement. For each flow cell, more than 200 beads were measured and measurements for each concentration were performed at least in duplicate. Data analysis was done as described previously[62].

For the calculation of the end-to-end distance, 25 beads around the fitted RMS of a population were selected and the 2.5% most distant positions of each bead were taken. The end-to-end distance was obtained by triangular calculation for each point and the resulting populations were fitted with a skewed Gaussian fit. A pairwise distribution was obtained by taking the difference between each point to all others and the resulting populations were fit with a Gaussian distribution.

**DNA-bridging assays**. The DNA used for the bridging assay is the same as that used for TPM and was $^{32}$P-labelled[63]. The DNA-bridging assay was performed as described previously[34,36] with minor modifications. Streptavidin-coated Magnetic M-280 Dynabeads (Invitrogen) were resuspended in buffer (20 mM Tris-HCl pH 8.0, 2 mM EDTA, 2 M NaCl, 2 mg/mL BSA (ac), 0.04% Tween20) containing 100 fmol biotinylated 47% GC DNA (685 bp) and incubated at 1000 rpm for 20 min at 25 °C in an Eppendorf Thermomixer with an Eppendorf Smartblock 1.5 mL.

The beads with associated DNA were washed twice before resuspension in the incubation buffer. Radioactive $^{32}$P-labelled DNA and unlabelled DNA were combined to maintain a constant (2 fmol/µl) concentration and a radioactive signal around 8000 cpm, and then added to each sample. Next, protein was added to initiate formation of bridged protein–DNA complexes. Incubation buffer, DNA buffer and protein buffer were designed in such a way to make a constant experimental buffer: 10 mM Tris-HCl, pH 7.5, 150 mM NaCl, 5 mM KCl, 5% v/v glycerol, 0.016% Tween20, 0.8 mg/ml acetylated BSA, 1 mM MgCl$_2$, 1 mM spermidine, 1 mM DTT, 5 mM EDTA. The samples were incubated for 20 min at 1000 rpm at 25 °C in an Eppendorf Thermomixer with an Eppendorf Smartblock™ 1.5 mL. After the incubation the beads were washed with the same experimental buffers once and then resuspended in counting buffer (10 mM Tris-HCl, pH 8.0, 1 mM EDTA, 200 mM NaCl, 0.2% SDS). The radioactive signal of DNA was quantified by liquid scintillation and was used for the calculation of protein–DNA recovery (%) based on a reference sample containing the same amount of labelled $^{32}$P 685 bp DNA used in each sample. All DNA-bridging experiments were performed at least in triplicate.

**MJ1647 chromatin reconstitution**. As DNA template for reconstituted chromatin samples, a ~0.5 kb region from the *M. jannaschii* RNAP operon was PCR-amplified using primers FW1522 and FW1523 (Supplementary Table 1). Overall, 20 µl samples contained 1 µg PCR product and 0.73 µg A3, 1.14 µg MJ1647 or 0.79 µg MJ1647ΔTM in N200 buffer. Samples were incubated at 37 °C for 15 min before the addition of 80 µl 1.25× MNase digestion buffer (25 mM Tris/HCl pH 8.0, 31.25 mM NaCl, 6.25 mM CaCl$_2$) containing 1, 3 or 10 units MNase (Thermo Fisher). Samples were further incubated at 37 °C for 5 min before the addition of 10 µl stop buffer (2% SDS, 0.1 M EDTA pH 8.0). After Phenol:Chloroform purification and ethanol precipitation, the samples were resolved on 3% agarose gels post-stained with ethidium bromide.

**MJ1647 cross-linking experiments**. In all, 10 µl samples contained 20 µM MJ1647 monomers and the indicated amounts of ~0.5 kb DNA (see above) in modified N(200) buffer with Tris being replaced by 10 mM HEPES/NaOH pH 7.5 as buffer system. Samples were incubated at 37 °C for 15 min before the addition of 5 mM freshly prepared BS$^3$ (Thermo Fisher). For cross-linking, samples were incubated at 25 °C for 40 min before the addition of SDS-PAGE loading dye containing Tris to quench the cross-linking reaction. Samples were resolved on 4–20% Mini-Protean TGX protein gels (Bio-Rad) and visualised by SYPRO-Orange staining (Thermo Fisher).

**MJ1647 crystallisation**. Crystallisation trials of purified MJ1647 were set up at a concentration of 3 mg/ml using commercial sparse-matrix screens. Crystal hits were identified in a condition containing 0.1 M NaOAc pH 4.6, 10% (w/v) PEG3350 and 5% (w/v) tacsimate pH 4.6. Crystals from this drop were cryoprotected in mother liquor (0.1 NaAC pH 4.6, 10% (w/v) PEG3350, 5% (w/v) tacsimate pH 4) supplemented with 12% PEG 400 as additional cryprotectant before being flash- frozen in liquid nitrogen. Data were collected to a resolution limit of 1.9 Å from a single flash-frozen crystal at 100 K using a Pilatus 6M-F detector on beamline I02 at the Diamond Light Source, UK. A total of 1200 frames of 0.15° rotation were recorded with 1 s exposure. Indexing and integration were carried out automatically using the DIALS pipeline and suggested that the crystal belongs to monoclinic space group P2 or P21. Assignment as P21 was confirmed

following scaling, and subsequent analysis of the reduced dataset with POINTLESS[64].

**MJ1647 crystal structure determination**. Phases were determined by molecular replacement using the structure of HMfA histone from *M. fervidus* (PDB entry 1B67). The space group and cell dimensions along with the monomeric molecular weight of MJ1647 suggested the presence of four chains in the asymmetric unit and a solvent content of 47%. Clear solutions for four molecules were obtained that clustered in the asymmetric unit as two dimeric pairs associated in a manner identical to that previously observed for classical histone dimers, confirming that the solution was essentially correct. Electron density maps calculated using phases from the assembly of four correctly positioned HMfA chains were of excellent quality and showed clear regions of additional density extending from the C-terminal regions. This starting model and the MJ1647 primary sequence were used as input for automated model building in Buccaneer[65] which produced a new set of coordinates comprising ~90% of the MJ1647 structure including additional helical segments corresponding to the C-terminal, non-conserved extension. Further manual model building in Coot[66] interspersed with crystallographic refinement using PHENIX[67] enabled an essentially complete model of two MJ1647 dimers to be built. The quality and completeness of the final model and final refinement statistics are summarised in Table 1.

**Structural modelling of MJ1647 dimers and tetramers**. For the AlphaFold predictions, we ran MMseqs2 and LocalColabFold on the high performance computing facility ALICE at Leiden University[38–41]. A multiple sequence alignment (MSA) for MJ1647 was generated with MMseqs2 (commit bfc6f85 from December 5 2021). Target databases used for this MSA were constructed by the ColabFold team (https://colabfold.mmseqs.com/) and include UniRef30, BFD, Mgnify, MetaEuk, SMAG, TOPAZ, MGV, GPD, and MetaClust2. The search-sensitive parameter was set to 8. The constructed MSA was used as an input for LocalColabFold (LocalColabFold: commit 6b76904 from December 4 2021, ColabFold: commit 33fcb9a from December 7 2021) to predict the dimer and tetramer structures of MJ1647. No templates and 3 recycles were used for these predictions. The structures were relaxed by AlphaFold's AMBER forcefield.

**Multi-round transcription assays**. The DNA template for in vitro transcription was constructed as follows: pGEM-T easy plasmid harbouring the strong T6 promoter[68] was modified by inserting a 500 bp region derived from the *rpo2* gene within the *M.jannaschii* genome (position 973045 to 973544, Supplementary Table 1) via *Nco*I (centred 66 nt downstream of the TSS) and a newly introduced *Bam*HI site downstream yielding a 574 nt run-off transcript when linearised with *Bam*HI. The design of in vitro transcription assays was based on[28,69,70]. Multi-round transcription samples contained 100 ng/µl *Bam*HI-linearised plasmid, 67 nM RNAP, 1.25 µM TBP, 0.13 µM TFB, 2.5 µM TFE, 500 µM ATP/GTP/CTP, 25 µM UTP supplemented with [α-$^{32}$P]-UTP and the indicated concentrations of histone dimers in modified HNME buffer (10 mM HEPES/NaOH pH 7.3, 250 mM NaCl, 2.5 mM MgCl$_2$, 0.1 mM EDTA, 5% glycerol, 6.7 mM DTT) supplemented with 67 µg/ml BSA and 5 µg/ml heparin. Samples were incubated for 15 min at 65 °C after which 8 µl were withdrawn and transferred into 200 µl stop mix (3.75 M NH$_4$-Acetate, 10 mM EDTA, 200 µg/ml GlycoBlue as co-precipitant (Thermo Fisher). Samples were purified once by acid-Phenol:-Chloroform (Thermo Fisher) extraction followed by Chloroform

extraction and ethanol precipitation. Pellets were washed twice with 70% ethanol, resuspended in 10 µl formamide loading dye (95% deionised formamide, 18 M EDTA, 0.025% SDS) and incubated for 5 min at 95 °C before loading onto an 8% polyacrylamide, 7 M Urea, 1× TBE sequencing gel. Transcripts were detected by phosphor imagery and quantification of bands was performed using the ImageQuant TL software (GE Life Sciences).

**Synchronised in vitro transcription assays**. The truncated TFB variant termed TFBcore that comprises only the C-terminal cyclin folds and is transcriptionally inactive[71] was used to outcompete full-length TFB and prevent transcription reinitiation[46]. It was produced and purified as described for full-length TFB[60]. Synchronised transcription samples were modified from the composition of multi-round transcription samples by changing the supplied nucleotides. Samples contained 100 ng/µl *Bam*HI-linearised plasmid, 67 nM RNAP, 1.25 µM TBP, 0.13 µM TFB, 2.5 µM TFE, 100 µM ATP/GTP and the indicated concentrations of histones in modified HNME buffer (10 mM HEPES/NaOH pH 7.3, 250 mM NaCl, 2.5 mM MgCl$_2$, 0.1 mM EDTA, 5% glycerol, 6.7 mM DTT) supplemented with 67 µg/ml BSA and 5 µg/ml heparin. Samples were incubated for 10 min at 65 °C to allow DNA chromatinization and PIC assembly. Transcription was initiated by the addition of 500 µM ATP/GTP/CTP, 25 µM UTP supplemented with [α-$^{32}$P]-UTP and 5 µM TFBcore (40-fold excess over TFB) to limit reinitiation.

**Statistics and reproducibility**. All experiments were performed at least in triplicates with the exception of TPM data that are from two to three replicates.

**Reporting summary**. Further information on research design is available in the Nature Portfolio Reporting Summary linked to this article.

## Data availability

The unedited/uncropped gels for all figures are included in Supplementary Figs. 10–20. The source data behind the graphs in Figure panels 6a and 6b are included in Supplementary Data 1 and 2, respectively. Sequence data and chromatin particle position frequency distribution files in bedGraph format are deposited at NCBI GEO under accession code GSE216101. Tethered particle motion and DNA-bridging data are stored at 4TU repository with https://doi.org/10.4121/20079704. X-ray structural data deposited in the PDB database with accession number 8BDK.

## Code availability

Analysis code for the MNase-seq data used to generate Fig. 5 and Supplementary Figs. 8 and 9 will be available from the authors upon request.

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

## Acknowledgements

Research in the RNAP lab at UCL ISMB was funded by a Wellcome Investigator Award in Science to Finn Werner, grant number WT 207446/Z/17/Z. Research in the lab of Remus Dame was funded by grants from the Netherlands Organisation for Scientific Research [VICI 016.160.613/533 and OCENW.GROOT.2019.012]. We thank Karen Moore and Konrad Paszkiewicz at the University of Exeter DNA Sequencing Service for technical help. Dominique Barir Jensen (UCL) helped with initial chromatin isolation experiments and Nikos Pinotsis (Birkbeck College) helped with SEC-MALS experiments.

## Author contributions

S.O., F.B., A.M., D.B., Z.S., T.F. and N.V. designed and performed experimental work. S.O., F.B., Z.S., S.S., K.S., D.M. and N.A.K. performed computational data analyses. N.A.K., K.T., R.T.D. and F.W. acquired funding, supervised and coordinated the work. F.B. and F.W. wrote the manuscript with input from all authors.

## Competing interests

The authors declare no competing interests.
