## [Peer Review File · Communications Biology]

Reviewers' comments:

Reviewer #1 (Remarks to the Author):

I would like to thank all authors for this interesting manuscript. In this work, the authors focus on a histone-fold (HF) containing protein found in the Methanococcales archaea.

The particularity of this protein is to associate a bona-fide histone fold domain to a lesser known but structured domain. Using a combination of carefully executed in-vitro experiments and some in-silico protein structure prediction, this manuscript demonstrates that this protein has the ability to bind DNA, and to mediate trans-interactions between HF dimers. In doing so, this histone paralogue can mediate the trans interaction of two DNA molecules -DNA bridging-. Further, the authors establish that this protein can compete with the transcription apparatus and repress transcription as efficiently as histone do. At last, nucleosome positioning / genome accessibility is probed by MNase. The observation of an histone paralogue that can mediate DNA bridging is new and in my opinion convincingly demonstrated.

There are two main points that I would like to discuss :

-Abundancy is a key element of NAPs function. However the paper doesn't mention MJ1647 abundancy relative to the other histone paralogues, considering that the author already make use of publicly available transcriptomic data, they could use this data to show how abundant MJ1647 is in comparison to other histones.

I could not find the reference of the data used for the classification of highly and lowly expressed genes shown on fig5C, but data from <https://doi.org/10.1128/AEM.00180-19> for example would be enough to provide a rough estimate of MJ1647 abundancy. I believe that knowing if MJ1647 is as abundant as the other paralogues could help on 2 fronts : it could provide clue to its function(global chromatin organiser vs more subtle role) and it could give a sense of how realistic it is to expect seeing the impact of MJ1647 by MNase-seq.

-I though that the MNase-seq experiment is technically sound and well presented. Unfortunately, I felt that its conclusions don't bring much to the story. Furthermore and as mentioned above, it's not clear to me that MNase is the technique of choice to probe the contribution of MJ1647 on genome structure / accessibility, particularly as we don't know how abundant the protein of interest is. While I realise the labour and the cost of such experiment, I believe that the manuscript would benefit from a radical shortening of the MNase-seq part (particularly l414 to 431).

minor points :

-The first paper describing MJ1647 properties (ref23, Li et al 2000) could perhaps be cited earlier in the manuscript (at line 95 for example). In addition, upon reading the ref23, I realised that previous finding described how MJ1647 forms dimers upon fixation. While I think that the results presented here convincingly demonstrate tetramerization, I think that this discrepancy should be discussed.

-l84, While histones are highly abundant in *T. kodakarensis*, they are not in *H. salinarium*. A global role of histone in *H. salinarium* is unlikely as Amy Schmidt lab has shown. Thus putting *T. kodakarensis* and *H. salinarium* side by side could be misleading.

-Figure3: The figure and the legend don't correspond : it seems that 3E is the insert mentioned in the legend of 3D and 3F the experiment mentioned in the legend as 3E.

While I understand the purpose of a 'line to guide the eye', this actually affects how the reader sees the data. In my opinion this is not necessary here, especially because all trends are very clear.

-Figure4: I found the figure in its current form a bit confusing. The zoom of fig4E could perhaps be associated to the fig3C/D ? In addition, I believe that a color legend could be added to fig2B. At last mentioning what's crystal and what AF2 would could be good. Fig4F: Ve should be spelled.

-line 366: I did not understand what technical limitation is referred to for the 100bp. After trimming and mapping many PE reads span less than 100bp as shown on fig5A.

-l446: As mentioned in the discussion the link between histone occupancy probed by MNase and RNA levels might not be causal, so it does not suggest a regulatory role. I guess 'consistent with a putative' might be more appropriate.

-l449: bacteria instead of "the second prokaryotic domain of life, the bacteria" suggest 3 domains of life (2 prokaryotes and eukaryotes), this is more than controversial so maybe shortening would avoid any unnecessary confusion.

-l618 to 624 : As mentioned above, I find that comparing *H. salinarium*, *H. volcanii* and *T. kodakarensis* is a bit misleading as histones have been shown to be lowly abundant in the two halophilic archaea. Also latest results on histone in *H. salinarium* explain how "HpyA functions as a direct activator of iron regulatory genes and a global indirect regulator of diverse pathways." (<https://doi.org/10.1093/nar/gkab1175>) hence the comparison with *M. jannashii* might not be relevant.

-I could not find the ref of the transcriptomic data used

With my best regards,

Antoine Hocher

Reviewer #2 (Remarks to the Author):

All eukaryotic, many archaeal species, and now some Bacteria encode the DNA binding proteins histones. Between domains, the core histone fold composed of three α -helices ($\alpha 1$, $\alpha 2$, $\alpha 3$) and two connecting loops (L1, L2) is highly conserved; however, eukaryotic histone structure includes N- and C-terminal extensions which are common targets for post translational modifications. On the contrary, there are only a handful of archaeal histones with any extensions (excluding halophilic-encoded histone with repeated sequences that although monomeric mimic a histone dimer). While eukaryotic and archaeal histones maintain the same DNA binding properties (in contrast with newly identified bacterial histone proteins) and bend the phosphodiester backbone around themselves at much the same angle, they diverge when it comes to higher order chromatin structure. The bulk of eukaryotic chromatin is characterized by distinct nucleosomes composed of an octamer of 4 obligate heterodimers wrapping 147 bp of DNA around it; most archaeal histones can form hetero- or homo-dimers and form long polymers wrapping anywhere from 60-500 bp of DNA in an extended superhelical structure termed a hypernucleosome.

Over 20 years ago, Li et al characterized a unique MJ1647 archaeal histone variant with a C-terminal extension, detailing thermostability, DNA binding and dimerization, the impact of retention or loss of the C-terminal extension on many properties, CD spectroscopy, and more. The "slinky" structure of archaeal histone-based chromatin was revealed only in 2017, and the current manuscript readdress this unique histone variant (MJ1647) in a new light.

Here, the authors establish (again) that MJ1647 is indeed a unique archaeal histone variant with DNA-

binding activity and argue for DNA-bridging activity; significant differences in reported details with literature established DNA binding and solution parameters complicate interpretations. TPM and structural studies provide evidence of DNA binding and compaction, likely through MJ1647 binding to 60 bp fragments. The crystal structure (well-resolved) displays a dimeric structure with the C-terminal pair of helices forming a tight monomer-monomer interface in the dimer; the structure of the dimer is real and matches all expectations for dimeric histone interfaces. Further, the dimeric structure fully accounts for the CTD (or TM, although likely improperly termed) helices. The authors report that MJ1647 forms obligate tetramers in solution, but then report a dimeric atomic structure. The results are confusing, and the tetrameric results that form the basis of the first half of the manuscript do not match previously reported results. The authors turn to modeling, that is unfortunately not supported by experimental data in this manuscript (disruption of predicted salt bridges in the hypothesized TM do not result in disrupting tetramers to dimers as predicted; E95K has essentially no impact, invalidating the double charge reversal, and K80E does not result in an obvious shift to a dimer), to hypothesize a tetramer when a tetramer is not resolved through direct observation.

Critical and important differences in the conclusions between these new results and previous results are not discussed throughout, the tetramer model is not supported, and many omissions/additional experiments must be remedied before publication may be possible. The binding mode of the any tetramer does not support a 60 bp footprint. Given the rather major concerns listed below, and incremental advances in other areas, publication cannot be recommended without major experimental and written revisions.

1. Histone proteins, albeit with a dramatically different oligomeric state and different mode of DNA binding, are now reported in Bacteria. The first line of the introduction thus requires additional clarification. Lines 450-452 should be reworked as well.

Given the recent report of the quaternary structure of eukaryotic telomeric chromatin structures (and the rather amazing match to the hypernucleosome structure of archaeal systems), the absolutes in lines 40-41 of the introduction necessitate change.

Lines 60 – 65 are hypotheses with potential truths but are currently not supported by experimental evidence; changes to the text are necessary to ensure this theory is presented as theory, not fact.

2. Line 66. Recent, and convincing evidence (doi: 10.1007/s00412-021-00759-8.), demonstrates that the 30 nm fiber is an artifact due to specific ionic concentration only found in vitro. It may be advantageous to explicitly state such.

3. Li et al used covalent crosslinking to determine the oligomeric state of MJ1647 in solution, determining that a mixture of monomers and dimers were prevalent in solution. These results from 2000 match the crystal structure, not the model of a tetramer. No indication of tetramers was reported in 2000. Deletion of the extra helix (termed the TM in the manuscript under review) did not impact dimerization in 2000; as neither MJ1647 variant resolved as a tetramer in Li et al, the impact of an identical deletion of the TM did not influence tetramerization in the 2000 study. This important difference is not discussed; revised versions must address this discrepancy. Although BS crosslinking and SEC are reported here (both supportive of tetramers), the differences with previous results are striking. Modeling, with only weak salt bridge reversal data, does not support a model of dimer-dimerization nor DNA binding to protect 60 bp fragments.

4. The proposed model of the MJ1647 tetrameric complex has the known DNA binding surfaces pointed 180° away from each other. Such a model does not permit binding of a single 60 bp continuous piece of DNA. The model proposed is completely incongruent with a model of binding

demonstrated in Fig 2B on a continuous piece of 60 bp DNA. There is no mechanical path to allow the model of a tetramer in 4C to bind a continuous 60 bp DNA. The proposed tetrameric model could bridge a long piece of DNA bent from one binding surface to the next or to permit capture two different DNA molecules with any length of intervening DNA. However, the model is not sufficiently supported to be believable.

5. The absence of mixed dimers (A3 and MJ1647) is perplexing given that no obvious structural clashes should preclude heterodimer formation. The defined crystal structure of the MJ1647 homodimer suggests that heterodimer formation would only result in "two-hanging CTD helices from a single monomer". Can the authors speculate as to why heterodimer formation would be "not observed"? What techniques were used to monitor heterodimer formation? Data not shown is not terribly useful.

5. EMSAs in Fig 2 A, B, C match those of Li et al, but Fig 2 G, H, I (with longer pieces of DNA (linear pBR322 and 5.5 kb fragments), the binding patterns of MJ1647 are rather dramatically different. Discussion is necessary to explain the radical differences in a nearly identical experiment.

6. Lines 254 and 282: The information is referencing figure 3F and not 3E. Additionally, there is no explanation of what 3E represents.

7. Bridging, as determined by retention of a second radiolabeled 30-mer on beads via MJ1647 bound to a biotinylated 30-mer, saturates at just ~50% recovery at ~100-fold (~2 μ M) higher concentrations than the RMS reductions (measured via TPM) at ~20 nM. At 20 nM, "bridging" is not observed, although TPM reductions in RMS argues for strong, cooperative, 60 bp DNA binding under near identical conditions. This discrepancy, of at least two orders of magnitude suggests, that one or the other assay is not effectively measuring DNA binding of MJ1647. As RMS decreases via TPM are well-established, it appears, to the naïve reader, that the bridging noted in Fig 3 F is not physiological and thus has no bearing on MJ1647 function. Discussion of this discrepancy is necessary in revisions. Bridging of a continuous piece of DNA that is just 60 bp is not possible based on the model of 4C.

8. MNase-seq of chromatin was restricted to minimally 100 bp, immediately eliminating identification of the preferred, continuous 60 bp fragments identified previously (Li et al), and here (Fig 2 and supplementary). This is a major concern. Given that many reports, including many from these authors, include smaller Mnase-resistant fragments, the lack of including such values is a major disappointment and limits interpretations. The MNase-seq data collected does not reveal anything about MJ1647 specifically (as the relative contribution of MJ1647 to overall chromatin structure may be very small) and only corroborates what was already known about archaeal histone-based chromatin. Would it be possible to repeat the MNase-seq with an antibody specific to this histone variant for IP (ChIP-seq – one could use the same or similar analysis pipeline from the MNase-seq experiment)? The potential to determine MJ1647 positions across the genome is great and would be incredibly informative, especially if comparing/contrasting the different cell cycle phases.

9. The complete absence of any 60 bp protection fragments in Mnase-seq further argues that the tetrameric model is not accurate and does not reflect a natural state of MJ1647 in vivo. Arguments made in the discussion that histone variants partition chromatin into canonical hypernucleosomes and MJ1647 filaments is not supported by any experimental data. The pattern observed is however completely expected for dimeric histone structures forming dynamic hypernucleosomes. Given the repeated and exhaustive nature of a dimer being the obvious state of MJ1647, it is perplexing why a non-physiological model that does not match the authors data is reported. Nothing about the Mnase-seq data in vivo supports a role for a tetrameric version of MJ1647 being physiological. The authors are surely aware of a defined genetic system of M.j., which could and should be used to i) delete

MJ1647, ii) introduce a TM mutant, and iii) introduce salt-bridge variants to aid an information regarding potential tetramers in vivo.

10. The in vitro transcription results exactly match previous results from Wilkinson et al, and thus represent only a minor incremental advance. The impact of the TMs, or bridging, or potential tetramerization are not resolved by such assays.

11. Lines 479 - 481: Chromatin in other archaeal systems does inhibit elongation (Sanders et al) (Line 594). As is, the statement seems much too broad and is in conflict with accepted knowledge about archaeal transcription. Perhaps a note here on how the histones tested (A3, MJ1647, MJ1647 Δ TM) showed no inhibition of transcription elongation unlike the effects seen in some other archaeal species might alleviate this contradiction.

Reviewer #3 (Remarks to the Author):

The authors characterize the structure and function of the histone paralogue MJ1647 from *M. jannaschii* archaea. This paralogue has a unique C-terminal extension differently to the majority of other archaeal histones. In this manuscript the authors very thoroughly characterize this histone, as well as a more typical short A3 histone paralogue. Moreover, they show a crystal structure of the MJ1647 dimer as well as AlphaFold2 models of the corresponding histone tetramers. This study shows for the first time that such a unique histone is able not only to bind DNA, but also to cross-bridge two DNA molecules in trans. The authors also performed experiments to test more large scale histone binding to the DNA, and tested the effects of histone binding on transcription. This study is very systematic and clean, the results are described very logically. It was a pleasure to read this paper. I recommend this paper to be published pretty much as is with 2 very minor corrections (see below).

- Figure 3: panel F is not described in the figure legend. And in the main text lines 283-284 reference should also be to Fig 3F and not Fig3E.

- Inconsistency with the resolution – should be 1.9 Å instead of 1.6 Å (line 514)

Overall comments.

Below we address in a point-to-point fashion all three reviewer's concerns. To assuage one of the reviewer's concerns about the oligomerization state of the histone variant MJ1647, we have carried out substantial additional experimentation including native mass spectrometry that prove beyond any reasonable doubt that this histone forms a tetramer (and not dimer) in solution. In addition, we have engineered double-cysteine variants that covalently link the monomers, an analysis that elegantly validates our structural model of the tetramer. Finally, our collaborators have recently prepared a manuscript that demonstrates a similar tetramerisation behaviour in other archaea (<https://biorxiv.org/cgi/content/short/2023.06.01.543357v1>) – making our manuscript the first one to describe this unorthodox property. In addition to the above, we have addressed all other concerns and followed the suggestions of the reviewers to improve our manuscript.

Reviewer #1 (Remarks to the Author):

I would like to thank all authors for this interesting manuscript. In this work, the authors focus on a histone-fold (HF) containing protein found in the Methanococcales archaea.

The particularity of this protein is to associate a bona-fide histone fold domain to a lesser known but structured domain. Using a combination of carefully executed in-vitro experiments and some in-silico protein structure prediction, this manuscript demonstrates that this protein has the ability to bind DNA, and to mediate trans-interactions between HF dimers. In doing so, this histone paralogue can mediate the trans interaction of two DNA molecules -DNA bridging-. Further, the authors establish that this protein can compete with the transcription apparatus and repress transcription as efficiently as histone do. At last, nucleosome positioning / genome accessibility is probed by MNase. The observation of an histone paralogue that can mediate DNA bridging is new and in my opinion convincingly demonstrated.

There are two main points that I would like to discuss:

-Abundancy is a key element of NAPs function. However the paper doesn't mention MJ1647 abundancy relative to the other histone paralogues, considering that the author already make use of publicly available transcriptomic data, they could use this data to show how abundant MJ1647 is in comparison to other histones.

I could not find the reference of the data used for the classification of highly and lowly expressed genes shown on fig5C, but data from <https://doi.org/10.1128/AEM.00180-19> for example would be enough to provide a rough estimate of MJ1647 abundancy. I believe that knowing if MJ1647 is as abundant as the other paralogues could help on 2 fronts: it could provide clue to its function(global chromatin organiser vs more subtle role) and it could give a sense of how realistic it is to expect seeing the impact of MJ1647 by MNase-seq.

We estimated the transcript abundance based on our recent RNA-seq data (Smollett et al., Nature Microbiology 2017) and added this information to the figure legend. We used the RNA-seq data to estimate mj1647 transcript abundance as suggested by the reviewer and included the following sentence in the manuscript”

‘Based on M. jannaschii RNA-seq data 27, mj1647 transcripts comprise ~4% of histone-encoding transcripts and show a similar expression level as the gene encoding Alba.’

-I thought that the MNase-seq experiment is technically sound and well presented. Unfortunately, I felt that its conclusions don't bring much to the story. Furthermore, and as mentioned above, it's not clear to me that MNase is the technique of choice to probe the contribution of MJ1647 on genome structure

/accessibility, particularly as we don't know how abundant the protein of interest is. While I realise the labour and the cost of such experiment, I believe that the manuscript would benefit from a radical shortening of the MNase-seq part (particularly l414 to 431).

We have followed the reviewer's suggestion and shortened the MNase-seq part. But we politely disagree that the nucleosome sequencing 'does not bring much to the story' – our *M. jannaschii* results are directly comparable to *T. kodakarensis*, and add to the body of evidence describing histone chromatinisation in euryarchaea. Whereas it is true that MNase-seq is not the ideal experiment to discriminate between A3 and MJ1647 binding in the genome, our extensive attempts to ChIP MJ1647 were unfortunately not successful, in contrast to ChIP-seq experiments for the *M. jannaschii* transcription machinery (including TBP, TFB, Spt4/5 and RNAP) which were highly efficient. This lack of histone *ChIPability* has been observed in other archaea and is likely because the DNA solenoid in the hypernucleosome obscures any epitopes required for the immunoprecipitation (ChIP).

MNase-seq results provide a valid approach to test *in vivo* chromatinization in *M. jannaschii* in its entirety, to assess the 'formatting' of the genome as guided by the plethora of chromatin proteins, including histones and other proteins such as Alba. We have analysed now the MNase-seq data with regards to the local bias of 120/180 bp over 90/150/210 bp fragments to identify possible MJ1647 binding sites that we expect to show 60bp increments (Supplemental Figure 9).

minor points

-The first paper describing MJ1647 properties (ref23, Li et al 2000) could perhaps be cited earlier in the manuscript (at line 95 for example). In addition, upon reading the ref23, I realised that previous finding described how MJ1647 forms dimers upon fixation. While I think that the results presented here convincingly demonstrate tetramerization, I think that this discrepancy should be discussed.

This point was also raised by reviewer 2 and we added text to discuss this as described in detail further below this document, including the correct citation.

-l84, While histones are highly abundant in *T. kodakarensis*, they are not in *H. salinarium*. A global role of histone in *H. salinarium* is unlikely as Amy Schmidt lab has shown. Thus putting *T. kodakarensis* and *H. salinarium* side by side could be misleading.

We agree that the different roles of histones in *T. kodakarensis* and *H. salinarum* argue for separation and additional explanations and revised the text as follows:

'Supporting this notion, the deletion or mutation of histones in Methanosarcina mazei and Thermooccus kodakarensis results in aberrant gene expression pattern^{14,15}, and a severe impairment of DNA recombination¹⁶. Notably, the borders between chromatin proteins and transcription regulators are fluid. In Halobacterium salinarum, the single histone HypA evolved to act as a transcription regulator that binds to discrete genomic sites rather than playing a role in genome compaction^{17,18}.'

-Figure3: The figure and the legend don't correspond : it seems that 3E is the insert mentioned in the legend of 3D and 3F the experiment mentioned in the legend as 3E.

Sorry! The figure legend has been corrected.

While I understand the purpose of a 'line to guide the eye', this actually affects how the reader sees the data. In my opinion this is not necessary here, especially because all trends are very clear.

We believe that these line lines do help the readers to better appreciate the trends as the plots are relatively busy. The lines do follow the data means very closely and do not change the interpretation of the data. For this reason, we prefer to keep them.

-Figure4: I found the figure in its current form a bit confusing. The zoom of fig4E could perhaps be associated to the fig3C/D ? In addition, I believe that a color legend could be added to fig2B. At last mentioning what's crystal and what AF2 would could be good. Fig4F: Ve should be spelled.

We modified Figure 4 as suggested by the reviewer.

-line 366: I did not understand what technical limitation is referred to for the 100bp. After trimming and mapping many PE reads span less than 100bp as shown on fig5A.

The technical limitation is due to the usage of QIAquick PCR purification protocol (Qiagen) that depletes dsDNA fragments < 100 bp. We have stated this now explicitly in the Material and Methods section. This does not mean that smaller fragments are completely absent, but they are underrepresented in the sequencing data. We have now introduced a sentence to clarify the technical limitation:

'For technical reasons, DNA fragments below 100 bp were depleted in the samples.'

-l446: As mentioned in the discussion the link between histone occupancy probed by MNase and RNA levels might not be causal, so it does not suggest a regulatory role. I guess 'consistent with a putative' might be more appropriate.

We have toned down this statement following the reviewer's suggestion. However, just because the causality of an interaction could work in both directions (e. g. chromatin blocks transcription, and/or transcription prevents chromatinization), this does not negate the causality, either way.

-l449: bacteria instead of "the second prokaryotic domain of life, the bacteria" suggest 3 domains of life (2 prokaryotics and eukaryotes), this is more than controversial so maybe shortening would avoid any unnecessary confusion.

The reviewer is correct that the dominating view in our field is that there are two primary domains (archaea and bacteria) and that eukaryotes form a sister group of the former. The sentence was altered as suggested:

'In bacteria, chromatin (or 'nucleoid') proteins such as E. coli H-NS occlude promoters but [...]

-l618 to 624 : As mentioned above, I find that comparing H. salinarium, H. volcanii and T. kodakarensis is a bit misleading as histones have been shown to be lowly abundant in the two halophilic archaea. Also latest results on histone in H. salinarium explain how "HpyA functions as a direct activator of iron regulatory genes and a global indirect regulator of diverse pathways." (<https://doi.org/10.1093/nar/gkab1175>) hence the comparison with M. jannashii might not be relevant.

We agree that the role of histones in Halobacterium might not be *directly* comparable to other euryarchaea and have modified the sentence accordingly leaving out the subclause referring to Halobacterium:

'Recent ATAC-seq data for H. volcanii rather point to an accessible promoter state independent of the transcription level of the gene⁵⁶.'

-I could not find the ref of the transcriptomic data used

The transcriptomic data were published in: *A global analysis of transcription reveals two modes of Spt4/5 recruitment to archaeal RNA polymerase.* Smollett K, Blombach F, Reichelt R, Thomm M, Werner F. *Nat Microbiol.* 2017 Mar 1;2:17021. doi: 10.1038/nmicrobiol.2017.21.PMID: 28248297. We have added the reference to the figure legend.

Reviewer #2 (Remarks to the Author):

All eukaryotic, many archaeal species, and now some Bacteria encode the DNA binding proteins histones. Between domains, the core histone fold composed of three α -helices ($\alpha 1$, $\alpha 2$, $\alpha 3$) and two connecting loops (L1, L2) is highly conserved; however, eukaryotic histone structure includes N- and C-terminal extensions which are common targets for post translational modifications. On the contrary, there are only a handful of archaeal histones with any extensions (excluding halophilic-encoded histone with repeated sequences that although monomeric mimic a histone dimer). While eukaryotic and archaeal histones maintain the same DNA binding properties (in contrast with newly identified bacterial histone proteins) and bend the phosphodiester backbone around themselves at much the same angle, they diverge when it comes to higher order chromatin structure. The bulk of eukaryotic chromatin is characterized by distinct nucleosomes composed of an octamer of 4 obligate heterodimers wrapping 147 bp of DNA around it; most archaeal histones can form hetero- or homo-dimers and form long polymers wrapping anywhere from 60-500 bp of DNA in an extended superhelical structure termed a hypernucleosome.

Over 20 years ago, Li et al characterized a unique MJ1647 archaeal histone variant with a C-terminal extension, detailing thermostability, DNA binding and dimerization, the impact of retention or loss of the C-terminal extension on many properties, CD spectroscopy, and more. The “slinky” structure of archaeal histone-based chromatin was revealed only in 2017, and the current manuscript readdress this unique histone variant (MJ1647) in a new light.

Here, the authors establish (again) that MJ1647 is indeed a unique archaeal histone variant with DNA-binding activity and argue for DNA-bridging activity; significant differences in reported details with literature established DNA binding and solution parameters complicate interpretations. TPM and structural studies provide evidence of DNA binding and compaction, likely through MJ1647 binding to 60 bp fragments. The crystal structure (well-resolved) displays a dimeric structure with the C-terminal pair of helices forming a tight monomer-monomer interface in the dimer; the structure of the dimer is real and matches all expectations for dimeric histone interfaces. Further, the dimeric structure fully accounts for the CTD (or TM, although likely improperly termed) helices. The authors report that MJ1647 forms obligate tetramers in solution, but then report a dimeric atomic structure. The results are confusing, and the tetrameric results that form the basis of the first half of the manuscript do not match previously reported results. The authors turn to modeling, that is unfortunately not supported by experimental data in this manuscript (disruption of predicted salt bridges in the hypothesized TM do not result in disrupting tetramers to dimers as predicted; E95K has essentially no impact, invalidating the double charge reversal, and K80E does not result in an obvious shift to a dimer), to hypothesize a tetramer when a tetramer is not resolved through direct observation. Critical and important differences in the conclusions between these new results and previous results are not discussed throughout, the tetramer model is not supported, and many omissions/additional experiments must be remedied before publication may be possible. The binding mode of the any tetramer does not support a 60 bp footprint. Given the rather major concerns listed below, and incremental advances in other areas, publication cannot be recommended without major experimental and written revisions.

1. Histone proteins, albeit with a dramatically different oligomeric state and different mode of DNA binding, are now reported in Bacteria. The first line of the introduction thus requires additional clarification. Lines 450-452 should be reworked as well.

We have included the most recent findings reported by Hocher *et al* in the revised version of the manuscript:

‘Histone-based chromatin is widely present in the Archaea and predates the origin of eukaryotes from an archaeal ancestor’^{1,2}. The histone fold itself predates the split of the archaeal and bacterial domains

³ and histone-based chromatin has recently been identified in the bacterium *Bdellovibrio bacteriovorus* albeit with a radically different DNA-binding mode compared to archaeoeukaryotic histones ⁴.’

In addition, following reviewer 1’s suggestion, we have introduced:

‘In bacteria, chromatin (or ‘nucleoid’) proteins such as E. coli H-NS occlude promoters but can also effectively stall transcription elongation complexes possibly by trapping them in topologically closed domains formed through DNA-bridging’.

Given the recent report of the quaternary structure of eukaryotic telomeric chromatin structures (and the rather amazing match to the hypernucleosome structure of archaeal systems), the absolutes in lines 40-41 of the introduction necessitate change.

To include the comparison to telomeric chromatin we have introduced the following sentence into the revised manuscript:

‘Eukaryotic histones H2A, H2B, H3 and H4 form well-characterized octameric nucleosomes that wrap ~147 bp of DNA. Canonical archaeal histones, in contrast, form dimers that assemble on DNA into ‘hypernucleosome’ particles of varying sizes with each dimer wrapping 30 bp of DNA ^{3,5}. The X-ray structure encompassing three Methanothermus fervidus histone homodimers bound to a 90 bp DNA fragment showed dimensions including the diameter and step size for each solenoid turn identical to those of the eukaryotic nucleosome. However, the archaeal hypernucleosome forms a rod-like protein core around which the DNA is wound in a left-handed solenoid ⁵. This arrangement is unlike the eukaryotic nucleosome octamer, but it shows similarity to the structure of chromatin formed by telomeric tetranucleosomes in eukaryotes ⁶.’

Lines 60 – 65 are hypotheses with potential truths but are currently not supported by experimental evidence; changes to the text are necessary to ensure this theory is presented as theory, not fact.

In order to meet reviewer-2’s suggestions to improve the stringency of the wording we have modified this section:

*‘The hypernucleosome size distribution in cells could be affected by different factors. Histone variants with weakened dimer-dimer interfaces might act as ‘capstones’ that limit hypernucleosome size ¹⁰. The number of stacking interactions available to different histone variants was shown to affect hypernucleosome stability in vitro ^{3,7}. Lastly, post-translational modification (acetylation) of histone lysines involved in hypernucleosome stacking interactions could be observed in *Thermococcus gammatolerans* and is likely to affect hypernucleosome stability in vivo¹¹.’*

2. Line 66. Recent, and convincing evidence (doi: 10.1007/s00412-021-00759-8.), demonstrates that the 30 nm fiber is an artifact due to specific ionic concentration only found in vitro. It may be advantageous to explicitly state such.

We notice the controversy around this subject and have modified the statement: *‘Eukaryotic nucleosomes self-interact to form phase separated aggregates or – under in vitro conditions – structured 30 nm fibres ¹⁵’.*

3. Li et al used covalent crosslinking to determine the oligomeric state of MJ1647 in solution, determining that a mixture of monomers and dimers were prevalent in solution. These results from 2000 match the crystal structure, not the model of a tetramer. No indication of tetramers was reported in 2000. Deletion of the extra helix (termed the TM in the manuscript under review) did not impact

dimerization in 2000; as neither MJ1647 variant resolved as a tetramer in Li et al, the impact of an identical deletion of the TM did not influence tetramerization in the 2000 study. This important difference is not discussed; revised versions must address this discrepancy. Although BS crosslinking and SEC are reported here (both supportive of tetramers), the differences with previous results are striking.

This concern is a key issue of reviewer-2 and we have spent considerable time and resources to address this query. Partially, there is a misunderstanding/misinterpretation, as protein cross-linking followed by SDS PAGE cannot reveal a 'mixture of monomers and dimers are prevalent in solution'. The monomeric state of histones is profoundly insoluble and only relevant under the denaturing conditions of SDS PAGE *in vitro* (and pre-dimerisation *in vivo*, although we assume that histones dimerise co-translationally). Truth to be told, **we are very sorry** not to have emphasised the differences between our findings and previously published results (Li et al., 2000) concerning the oligomerisation properties of MJ1647. We do this comparison justice in the revised manuscript and include additional evidence for the presence of tetrameric MJ1647 species based on native mass spectrometry. In conjunction with our SEC MALS results this leaves little room for doubt that MJ1647 forms tetramers. SEC-MALS is the method considered *Gold standard* in determining protein oligomerisation under native conditions and that technique has the advantage of being a sensitive yet robust solution/equilibrium method.

Formaldehyde cross-linking experiments (as used by Li et al. 2000) are less than ideal to determine the oligomerisation state of proteins because formaldehyde only cross-links amino acid side chains (predominantly lysine residues) that are in very close proximity ($\sim 2 \text{ \AA}$). In Li et al., the absence of evidence is not evidence of absence (of tetramerization), but a limitation of formaldehyde cross linking to detect tetramerisation. In essence, we disagree with the reviewer in the interpretation of (incomplete) cross-linking as an indication that MJ1647 reflects an equilibrium of dimers and monomers. Firstly, cross-linking is *per se* not an equilibrium method, but a method known to at times overemphasise poorly populated transition states unduly, but that's not the problem here. Secondly, the degree of cross-linking is influenced by the abundance of suitable cross-linking residue pairs. Our SEC-MALS (and SEC) data clearly demonstrate that MJ1647 under the experimental conditions is nearly monodisperse in solution, there is one dominant species: the tetramer. Both the elution volume (SEC) and multi-angle light scattering are consistent with a molecular weight of $\sim 44 \text{ kDa}$ predicted for a MJ1647 tetramer.

Additional evidence for tetramerization is provided by cross-linking using BS³, an better cross-linker with an effective cross-linking distance of 11.4 \AA , which is therefore less dependent on closely spaced lysine pairs. Our BS³ results show cross-linked 1647 species corresponding to dimers, trimers and tetramers – as would be predicted from a native tetrameric 1647 species.

We are a bit uncertain what the reviewer means with the following concerns: 'Deletion of the extra helix (termed the TM in the manuscript under review) did not impact dimerization in 2000'. We agree, as this is what our results show, i. e. that 1647 Δ TM forms stable dimers. Furthermore, '[...] the impact of an identical deletion of the TM did not influence tetramerization in the 2000 study'. As Li et al. did not detect any tetramers using formaldehyde cross linking, the deletion of the TM cannot be expected to alter the cross-linking pattern in their experiments.

To validate the tetramerization of MJ1647, we now corroborated our data even further with additional native mass spectrometry data (shown in Figure 1 panel h, i and j), which are in agreement with the SEC-MALS, SEC, and BS³ cross-linking data: A3 forms dimers, 1647 forms tetramers, and 1647 Δ TM forms dimers. All experiments in aggregate, there remains little room for doubt that MJ1647 does form tetramers in solution.

To further clarify the difference between crosslinkers, we added the following sentence:

'Previous formaldehyde cross-linking experiments retrieved only MJ1647 dimers, and not tetramers. The BS³ crosslinking with a longer cross-linking distance (~11.4 Å) compared to formaldehyde (~2 Å) clearly shows that MJ1647 forms tetramers (Figure 1h).'

Modeling, with only weak salt bridge reversal data, does not support a model of dimer-dimerization nor DNA binding to protect 60 bp fragments.

We beg to differ with the reviewer-2 in several ways. AI-guided modelling using AlphaFold2 plays an increasingly important role in the characterisation of molecular structures which are refractive to *bona fide* structure determination e. g. by X-ray crystallography. In our case, the protein crystallised exclusively in one oligomeric state but not another, a phenomenon that is not unusual when crystallising proteins, as the protein structure in the crystal lattice can interfere with protein-protein contacts, i. e. the *shape* of oligomer. But importantly, our *in silico* model of the MJ1647 tetramer is not an weird outlier as other archaeal histone variants are predicted to form tetramers via C-terminal extensions (<https://biorxiv.org/cgi/content/short/2023.06.01.543357v1>).

Having said that, we completely agree with the reviewer that computational models call for experimental validation. Our model predicts that a network of hydrophobic interactions and four salt bridges which stabilise the tetramer. With this combination of interaction, only the salt bridges are experimentally feasible for a rigorous mutational perturbation to corroborate the tetramer model. More importantly, while the salt bridges contribute to tetramerization, we don't claim that they are strictly necessary (or sufficient) for tetramer formation – the hydrophobic interaction network between dimers could go a long way in keeping the tetramer intact – this is not in conflict with our AF2 tetramer model. The spatial and chemical environment experienced by both amino acid chains constituting the salt bridge, K80 and E95, differs. While K80E has a strong effect on tetramerization under our experimental conditions, E95K does not. The negative charge of the K80E residue may perturb the TM structure in a different way than the positive charge of the E95K residue. This incomplete disruption of the tetramer by E95K is not unexpected.

To provide additional evidence for tetramerization according to the AF2 model, we probed the proximity of K80 and E95 by introducing cysteine pairs at positions 80 and 95. We first removed the two cysteine residues present in MJ1647 (C28S/C62S) and subsequently introduced K80C and E95C substitutions. All mutant variants were expressed at high levels, soluble and thermostable, and eluted in SEC similar to wild type MJ1647 consistent with a tetrameric state. The K80C mutation caused a slight destabilisation of the tetramers similar to K80E while the K80C/E95C double mutation was monodisperse and eluted similar to the C28S/C62S (Supplementary figure 5). The predicted C-alpha distance (7.7 Å) between residue 80 and 95 in the AF2 tetramer model is within the range of disulfide bonds of 3.5-7 Å (Figure 4C, annotated as chain B-D and A-C). By comparison, the distance between positions 80 and 95 of monomers within the dimer of the crystal structure (chains A-B and C-D) is too long (11.1 Å) and therefore incompatible with disulfide bond formation. Figure 4E shows the SDS PAGE analyses of the double cysteine substitution at three different reducing agent concentrations (1, 5 and 25 mM TCEP), compared to both wild type and the two single cysteine substitutions used as negative controls. The SDS PAGE reveals the double-cysteine variant as covalently linked dimeric MJ1647 species, a result that according to the distance information discussed above is only compatible with the predicted model of the tetramer and which runs as a dimer under the denaturing conditions of SDS PAGE (Figure 4E). This covalent dimer is very stable and resistant to 1 and 5 mM TCEP, and it requires high concentrations (25 mM) to reduce the disulfide bond to generate monomeric MJ1647. One of the

single cysteine negative control substitutions, E95C, shows a very weak covalent dimer band at low (1 mM) TCEP which is effectively reduced at intermediate (5 mM) TCEP concentrations, indicating a nonspecific association between surface exposed cysteine residues between non-associated proteins in solution. Neither wild type nor K80C variants show any covalent dimers.

For the discussion of the 60 bp DNA binding, please see point 4 below.

4. The proposed model of the MJ1647 tetrameric complex has the known DNA binding surfaces pointed 180° away from each other. Such a model does not permit binding of a single 60 bp continuous piece of DNA. The model proposed is completely incongruent with a model of binding demonstrated in Fig 2B on a continuous piece of 60 bp DNA. There is no mechanical path to allow the model of a tetramer in 4C to bind a continuous 60 bp DNA. The proposed tetrameric model could bridge a long piece of DNA bent from one binding surface to the next or to permit capture two different DNA molecules with any length of intervening DNA. However, the model is not sufficiently supported to be believable.

Sure, and we discussed the discrepancy in the discussion section of our manuscript. We agree with the reviewer that the tetramer model is not compatible with the canonical 30bp DNA per dimer binding mode of archaeal histones, which we highlight on page 18 (l. 538-547) in the original submission:

'Taken together, this suggests that the A3 and MJ1647 histones utilise two distinct DNA binding modes'.

We subsequently point out the varying degrees of confidence in different parts of the model and have revised the Discussion section to further highlight this outstanding question, now also including other alternative binding modes of histones in bacteria:

*'The tetrameric AF2 model of MJ1647 shown in figure 4 places the histone folds of the two dimers at diametrically opposing ends of the molecule, in agreement with the DNA-bridging property of MJ1647. This model cannot fully account for the 60 bp protection of DNA if the histone fold were to interact with the DNA in the same mode as canonical histones like A3. We surmise that MJ1647 tetramers use an alternative DNA binding mode compared to the canonical histone-DNA interactions observed in A3 and the dimeric MJ1647 Δ TM mutant. Alternative DNA-interaction modes among divergent paralogues of DNA-binding proteins are not unknown, and a highly unusual DNA-binding mode for a histone has indeed recently been reported in the bacteria *Bdellovibrio bacteriovorus* and *Leptospira interrogans*⁴. Furthermore, the 60 bp protection pattern could reflect other, possibly higher-order oligomers made of MJ1647 tetramers occurring in the presence of long DNA fragments used in the chromatin reconstitution experiments (Figure 2) and TPM analyses (Figure 3). This hypothesis is supported experimentally by the formation of higher-order oligomers which were strictly dependent on the presence of long DNA templates in BS³ cross-linking experiments (Figure 1i).'*

5. The absence of mixed dimers (A3 and MJ1647) is perplexing given that no obvious structural clashes should preclude heterodimer formation. The defined crystal structure of the MJ1647 homodimer suggests that heterodimer formation would only result in “two-hanging CTD helices from a single monomer”. Can the authors speculate as to why heterodimer formation would be “not observed”? What techniques were used to monitor heterodimer formation? Data no shown is not terrible useful.

The inability of A3 and MJ1647 to form mixed oligomers is not unexpected due to their high sequence divergence, see Figure 1A; it is not a merely a question of sterical clashes. The question is interesting, and we have carried out and included additional experiments based on co-expression of A3 and MJ1647 in *E. coli*. SEC analyses clearly demonstrate that the co-expressed A3 and MJ1647 elute in a fashion that is undistinguishable from individually expressed A3 and MJ1647, i. e. with elution profiles matching dimeric A3 and tetrameric MJ1647, respectively. We could not detect any elution behaviour compatible

with MJ1647/A3 heterodimers (Supplementary figure 1). In addition, we carried out BS³ cross-linking experiments using the purified, mixed MJ1647/A3 preparations. The cross-linking pattern of the mixed A3-1647 was in perfect agreement with the cross-linking pattern generated by the individual A3 and 1647 factors and without any evidence of heterodimeric species (Supplementary figure 1H). While our experiments cannot rule out a small fraction of heterodimers, they show a very strong preference for homo-oligomerisation.

We have introduced the following paragraph to discuss these results:

'We tested whether A3 and 1647 could form mixed oligomers by co-expressing A3 and MJ1647 in E. coli and analysing the complexes by SEC (Figure 1h). A3 and MJ1647 eluted from the SEC column in two sharp and distinct peaks which correspond to A3 dimers and MJ1647 tetramers according to their apparent molecular weight. This profile mirrored the individually expressed A3 and MJ1647 histones. We could not observe a peak in an intermediate elution volume predicted for A3-MJ1647 heterodimers (Supplementary Figure 1). In addition, we carried out cross-linking experiments with the amine-specific crosslinker bis(sulfosuccinimidyl)suberate (BS³) and analysed the cross-linked species by SDS-PAGE that provides increased resolution compared to SEC. A3 alone resulted in crosslinked dimers. MJ1647 alone resulted in crosslinked dimers, trimers and tetramers. The co-expressed A3 and MJ1647 led to crosslinking products reflecting a combination of A3 and MJ1647 but not new heterodimeric species, predicted to form an additional band between A3 dimers and MJ1647 dimers (Figure 1h).'

5. EMSAs in Fig 2 A, B, C match those of Li et al, but Fig 2 G, H, I (with longer pieces of DNA (linear pBR322 and 5.5 kb fragments), the binding patterns of MJ1647 are rather dramatically different. Discussion is necessary to explain the radical differences in a nearly identical experiment.

The key conclusion of these EMSAs – DNA compaction by MJ1647 – is the same in both articles. The binding patterns only differ in terms of the stability of the DNA-protein complexes. Whereas Li et al describe unstable complexes formed by MJ1647 (reflected in the smeared shifts of 'rMJ1647'), we observed stable MJ1647 binding that is accompanied by sharp bands. It would be interesting to compare the experiments in terms of binding buffer components, incubation temperature, complex assembly protocols, as well as gel buffer for the electrophoresis. All these parameters influence the appearance of EMSA signals. Unfortunately, neither Li et al nor the paper referenced therein (Sandman et al 1990 PNAS) include detailed experimental methods beyond the agarose concentration and V/cm used which would have allowed us to speculate about the instability of the 'rMJ1647'-pBR322 complexes observed in the Li et al. article.

6. Lines 254 and 282: The information is referencing figure 3F and not 3E. Additionally, there is no explanation of what 3E represents.

We are sorry for the oversight, the legend for Figure 3F was included in 3E, and this mistake has now been corrected.

7. Bridging, as determined by retention of a second radiolabeled 30-mer on beads via MJ1647 bound to a biotinylated 30-mer, saturates at just ~50% recovery at ~100-fold (~2 μM) higher concentrations than the RMS reductions (measured via TPM) at ~20 nM. At 20 nM, "bridging" is not observed, although TPM reductions in RMS argues for strong, cooperative, 60 bp DNA binding under near identical conditions. This discrepancy, of at least two orders of magnitude suggests, that one or the other assay is not effectively measuring DNA binding of MJ1647. As RMS decreases via TPM are well-established, it appears, to the naïve reader, that the bridging noted in Fig 3 F is not physiological and thus has no bearing on MJ1647 function. Discussion of this discrepancy is necessary in revisions. Bridging of a continuous piece of DNA that is just 60 bp is not possible based on the model of 4C.

The exact composition of the archaeal cytoplasm that influences protein-protein and protein-DNA interactions is poorly understood and binding *in vivo* may differ from the binding characteristics measured *in vitro*. Unfortunately even different *in vitro* methods can generate results that are not perfectly congruent (see point 5 above). However, the reviewer rightly points out that the differences in the MJ1647 concentration dependence between the two assays should be discussed in the body text. The 685 bp DNA we used here (not 30 bp as reviewer-2 states above) was at 5.2 nM in the bridging assay. Assuming 30 bp or 60 bp binding sites for MJ1647, that results in a binding site concentration of ~120 nM or 60 nM, respectively, explaining to some extent why higher MJ1647 concentrations are required. For comparison, for the single-molecule conditions of the TPM assays at least 10x lower DNA concentrations are used (100-500 pM). As this effect has been observed and published previously for many other chromatin proteins including H-NS, Rok, and MvaT that are well-known DNA bridgers, this is not a persuasive argument to doubt the physiological relevance of MJ1647 DNA-bridging. To further clarify this point we revised the text as follows:

The results show that MJ1647 is capable of bridging two DNA molecules in a fashion that is dependent on the TM, as neither A3 nor MJ1647 Δ TM are capable of bridging DNA (Figure 3E). Negative controls with MJ1647, but without biotin-labelled DNA, did not recover significant levels of ³²P-labelled DNA. The MJ1647 monomer concentration with a half-maximal recovery was approximately 2.5 μ M which is higher than the concentrations required for DNA compaction in TPM assays under single-molecule conditions. In part, this can be explained by the higher DNA concentrations used in the bridging assay that require a higher MJ1647 concentration to achieve saturation. Moreover, the general effect has been observed previously for other chromatin proteins known to bridge DNA such as H-NS, Rok, and MvaT^{33,34,36}.

Eventhough we take every possible precaution by using reaction vessels, pipette tips etc. made of 'low binding' materials, we cannot completely rule out that non-specific absorption of the protein contributes to this difference.

8. MNase-seq of chromatin was restricted to minimally 100 bp, immediately eliminating identification of the preferred, continuous 60 bp fragments identified previously (Li et al), and here (Fig 2 and supplementary). This is a major concern. Given that many reports, including many from these authors, include smaller MNase-resistant fragments, the lack of including such values is a major disappointment and limits interpretations. The MNase-seq data collected does not reveal anything about MJ1647 specifically (as the relative contribution of MJ1647 to overall chromatin structure may be very small) and only corroborates what was already know about archaeal histone-based chromatin. Would it be possible to repeat the MNase-seq with an antibody specific to this histone variant for IP (ChIP-seq – one could use the same or similar analysis pipeline from the MNase-seq experiment)? The potential to determine MJ1647 positions across the genome is great and would be incredibly informative, especially if comparing/contrasting the different cell cycle phases.

See also comment to reviewer-1. Our work on *M. jannaschii* chromatin started with nucleosome sequencing and data analysis in collaboration with Prof Nick Kent. We posit that MNase-seq provides a valid approach to characterise chromatinization in *M. jannaschii* in its entirety, to assess the *in vivo* footprint of all chromatin proteins in aggregate, and to establish the association between this chromatin footprint and mRNA levels. These objectives were achieved with our experiments, and the results show that *M. jannaschii* chromatin is directly comparable to *T. kodakarensis*, and therefore our results add to the body of evidence describing chromatin in euryarchaea. But we agree with the reviewer that the MNase-seq conditions are not the ideal experiments to discriminate between A3 and MJ1647 binding across the genome. Unfortunately, our extensive attempts to carry out ChIP-seq using

A3 and MJ1647 polyclonal antibodies were unsuccessful, in contrast to ChIP-seq experiments for the *M. jannaschii* transcription machinery (general factors TBP, TFB, Spt4/5 and RNAP) which were very efficient. This lack of histone 'ChIP-ability' has been observed in other archaea and is likely due to the fact that the DNA solenoid in the hypernucleosomes conceals any epitopes required for the immunoprecipitation.

9. The complete absence of any 60 bp protection fragments in Mnase-seq further argues that the tetrameric model is not accurate and does not reflect a natural state of MJ1647 in vivo. Arguments made in the discussion that histone variants partitions chromatin into canonical hypernucleosomes and MJ1647 filaments is not supported by any experimental data. The pattern observed is however completely expected for dimeric histone structures forming dynamic hypernucleosomes. Given the repeated and exhaustive nature of a dimer being the obvious state of MJ1647, it is perplexing why a non-physiological model that does not match the authors data is reported. Nothing about the Mnase-seq data in vivo supports a role for a tetrameric version of MJ1647 being physiological. The authors are surely aware of a defined genetic system of M.j., which could and should be used to i) delete MJ1647, ii) introduce a TM mutant, and iii) introduce salt-bridge variants to aid an information regarding potential tetramers in vivo.

We are sorry to disagree with the reviewer, see also our comments to query 8, there is simply no evidence to support reviewer-2's assumption of the '...repeated and exhaustive nature of a dimer being the obvious state of MJ1647...'. The combination of our state-of-the-art methods to characterise the oligomerisation state of MJ1647 (including SEC MALS and native Mass spectrometry) prove beyond any reasonable doubt that this protein is tetrameric in solution.

While it in theory would be possible to develop new experimental parameters that would allow an MNase-seq mapping to extend the size range down to 60 bp, this would be beyond the scope of the current manuscript. Please note that any larger MJ1647-fragmentation pattern, i. e. multiples of 60 bp, would be masked by the dominant 30 bp pattern generated by A3-like histones. The only additional piece of evidence of this revised MNase-seq protocols would be the actual 60bp DNA fragment – which we *already observe* in the reconstituted chromatin MNase digest shown in Figure 2E.

10. The in vitro transcription results exactly match previous results from Wilkinson et al, and thus represent only a minor incremental advance. The impact of the TMs, or bridging, or potential tetramerization are not resolved by such assays.

The *in vitro* transcription experiments applied in the Wilkins et al. paper are very different from the assays we have developed for this study. We discuss our methods and the results they generate in comparison to Wilkinson et al. in our manuscript. In short: We utilize two assays able to discriminate between single-round and multiple-round transcription, this is required to deconvolute inhibitory effects on transcription elongation and initiation, respectively:

- We show for the first time evidence for the effect of MJ1647 (the focus of our paper) on transcription (Wilkins describes the canonical A1, 2, 3 and 4)
- Wilkinson et al used multi-round transcription assays. We provide here the first insight into the effect of histone chromatinization on transcription elongation in *M. jannaschii* using synchronized transcription assays.

11. Lines 479 - 481: Chromatin in other archaeal systems does inhibit elongation (Sanders et al) (Line 594). As is, the statement seems much too broad and conflicts with accepted knowledge about archaeal transcription. Perhaps a note here on how the histones tested (A3, MJ1647, MJ1647ΔTM) showed no inhibition of transcription elongation unlike the effects seen in some other archaeal species might alleviate this contradiction.

We disagree with the reviewer that inhibition of transcription elongation by archaeal histones is “accepted knowledge”. First, there are no studies of transcription dynamics and the effect of histone chromatinization *in vivo*, making it very tentative to draw such a conclusion. Secondly, the Sanders et al. study, as well as Xie et al (2004), observed strong RNAP elongation complex pausing on an artificial template bearing a SELEX-derived high-affinity sequence as histone binding site. But such high-affinity ‘Widom-601’ like sites are not physiologically/biologically relevant nor representative. To overcome this limitation, we used a ‘native transcribed’ sequence, a 500 bp region derived from the *M. jannaschii* genome instead (rpo5 operon) which reflects true biology.

To meet the request of reviewer-2, we have rephrased the sentence to clarify that it refers to our results and their interpretation:

‘We concluded that the inhibitory effect of histone chromatinization on M. jannaschii transcription in vitro is largely due to interference with transcription initiation.’

Reviewer #3 (Remarks to the Author):

The authors characterize the structure and function of the histone paralogue MJ1647 from *M. jannaschii* archaea. This paralogue has a unique C-terminal extension differently to the majority of other archaeal histones. In this manuscript the authors very thoroughly characterize this histone, as well as a more typical short A3 histone paralogue. Moreover, they show a crystal structure of the MJ1647 dimer as well as AlphaFold2 models of the corresponding histone tetramers. This study shows for the first time that such a unique histone is able not only to bind DNA, but also to cross-bridge two DNA molecules in trans. The authors also performed experiments to test more large scale histone binding to the DNA, and tested the effects of histone binding on transcription.

This study is very systematic and clean, the results are described very logically. It was a pleasure to read this paper. I recommend this paper to be published pretty much as is with 2 very minor corrections (see below).

- Figure 3: panel F is not described in the figure legend. And in the main text lines 283-284 reference should also be to Fig 3F and not Fig3E.

Corrected

- Inconsistency with the resolution – should be 1.9 Å instead of 1.6 Å (line 514)

Corrected

REVIEWERS' COMMENTS:

Reviewer #1 (Remarks to the Author):

I have read all the authors comments and globally agree with them. I thank the authors for an interesting read.

Reviewer #2 (Remarks to the Author):

The authors provided detailed responses to each major concern from all reviewers and made substantial changes to the manuscript that clarify conclusions and further support the main conclusions from the manuscript. The improvements and efforts involved are notable, and while some disagreements in interpretations and biological significance likely remain, the revised manuscript is more compelling, more carefully interpreted, and further supported by important additional experimentation that was added to the manuscript. Several points regarding the tetramerization of MJ1647, including differences with results from Li et al, are now more carefully considered, compared and contrasted, and rationalized. This second review will therefore focus primarily only on the DNA binding mechanism of the tetrameric MJ1647. Important clarifications made to the text and additional experiment suffice to counter the primary concerns raised with the original manuscript.

Formaldehyde cross linking has limitations that may have precluded recovery of solution-based tetramers of MJ1647 in previous studies. The improved and additional data is supportive of a tetrameric structure in solution, but some concerns remain with one of the two proposed modes of DNA binding by the tetramer of MJ1647. While the modeled structure of the tetrameric species is high confidence and fully supportive of the DNA binding mode supporting DNA bridging, the model is NOT supportive of the binding of a tetrameric MJ1647 to a continuous 60 bp DNA. The model - that remains unchanged and singular - and that places the DNA binding surfaces at 180° opposed to each other, remains inconsistent with DNA binding to 60 bp DNA binding UNLESS i) the tetramer undergoes substantial rearrangements from the proposed model of the MJ1647 tetramer dependent on interactions between the TMs of each MJ1647 monomer or ii) a new mechanism of histone-DNA interaction is identified. The strong binding of MJ1647 to just 60 bp DNAs, shown both in previous manuscripts and here, argues for inclusion of a rationalized, second model of the tetramer of MJ1647 that differs from the sole model proposed in the revised manuscript that is supportive of i) rearrangements moving the DNA binding motifs AWAY from the 180° orientation of the tetramer model that is proposed, inclusive of how the TM domain interactions must be altered or can be retained with the current model or ii) that provides a plausible path of the bound DNA on the existing MJ1647 tetramer model supportive of binding to a 60 bp DNA.

The revised manuscript does provide a response to this remaining concern. Lines 907-930 now provide a short discussion of potential mechanisms supportive of alternative DNA binding modes of MJ1647. While future studies may identify the molecular basis of alternative modes of histone-DNA interactions, for MJ1647 or other histone or histone-like proteins, the current manuscript would be improved by inclusion of a model supportive of 60 bp DNA binding.

Finally, it is important to acknowledge that my initial comments were often blunt, and received with some trepidation by the authors, and apologies are necessary for such. The inclusion of a model supporting alternative modes of DNA binding would resolve all remaining concerns.